Eco-morphological diversity of larvae of soldier flies and their closest relatives in deep time

http://orcid.org/0000-0003-1893-3215 Baranov Viktor A. 1 baranow@biologie.uni-muenchen.de
Wang Yinan 2
Gašparič Rok 3
Wedmann Sonja 4
http://orcid.org/0000-0001-8254-8472 Haug Joachim T. 1 5
1 Biology II, Ludwig-Maximilians-Universität München , Planegg-Martinsried, Bayern , Germany
2 Association of Applied Paleontological Sciences , Logan, UT , USA
3 Oertijdmuseum, Boxtel, The Netherlands
4 Department of Messel Research and Mammalogy , Senckenberg Research Institute and Natural History Museum, Frankfurt/M. , Germany
5 GeoBio-Center, Ludwig-Maximilians-Universität München , Munich, Bayern , Germany
Tatarinova Tatiana
Electronic publication date: 2020 Nov 27
Publication date: 2020
Volume: 8
Electronic Location ID: e10356
Received 2019 Dec 21; Accepted 2020 Oct 23
Copyright: © 2020 Baranov et al.
Copyright year: 2020
Copyright holder: Baranov et al.
License: This is an open access article distributed under the terms of the Creative Commons Attribution License, which permits unrestricted use, distribution, reproduction and adaptation in any medium and for any purpose provided that it is properly attributed. For attribution, the original author(s), title, publication source (PeerJ) and either DOI or URL of the article must be cited.
License URL: https://creativecommons.org/licenses/by/4.0/

Keywords: Stratiomyomorpha, Diptera, Amber, Messel, Morphospace, Soldier Flies

Funding: Volkswagen Foundation LMU Junior Researcher Fund This project is supported by the Volkswagen Foundation in the frame of a Lichtenberg Professorship of Joachim T. Haug (Joachim T. Haug; Viktor Baranov). Viktor Baranov was supported by the LMU Excellence Initiative, via LMU Junior Researcher Fund. The funders had no role in study design, data collection and analysis, decision to publish, or preparation of the manuscript.

==============================
Stratiomyomorpha (soldier flies and allies) is an ingroup of Diptera, with a fossil record stretching back to the Early Cretaceous (the Barremian, about 125 MYA). Stratiomyomorpha includes at least 3,000 species in the modern fauna, with many species being crucial for ecosystem functions, especially as saprophages. Larvae of many stratiomyomorphans are especially important as scavengers and saproxyls in modern ecosystems. Yet, fossil larvae of the group are extremely scarce. Here we present 23 new records of fossil stratiomyomorphan larvae, representing six discrete morphotypes. Specimens originate from Cretaceous amber from Myanmar, Eocene Baltic amber, Miocene Dominican amber, and compression fossils from the Eocene of Messel (Germany) and the Miocene of Slovenia. We discuss the implications of these new records for our understanding of stratiomyomorphan ecomorphology in deep time as well as their palaeoecology.

Introduction

Stratiomyomorpha is a group of flies (Diptera), which includes more than 3,000 species of soldier flies and allies in the modern-day fauna (Pape, Blagoderov & Mostovski, 2011). The major ingroups of Stratiomyomorpha include: (1) Stratiomyidae, the group of true soldier flies, (2) Xylomyidae, the group of wood soldier flies and (3) Pantophthalmidae, the group of giant timber flies (Marshall, 2012). The group Stratiomyomorpha has a fossil record reaching back about 125 million years into the past, to the Barremian (Lower Cretaceous; Whalley & Jarzembowski, 1985; Mostovski, 1998). A now-extinct group of flies with long proboscides (Zhangosolvidae) from the Cretaceous has also been interpreted as an ingroup of Stratiomyomorpha (Peñalver et al., 2015).

Representatives of Stratiomyomorpha are widespread in modern ecosystems and diverse in their biology (Woodley & Thompson, 2001; Marshall, 2012). Larvae of different ingroups of Stratiomyomorpha vary in habitat preferences. Fully aquatic larvae are known in Stratiomyinae, Rhaphiocerinae etc. (ingroups of Stratiomyidae); other larvae develop in the terrestrial habitats as in the groups Pachygastrinae, Clitellariinae, Sargiinae or Hermetiinae (ingroups of Stratiomyidae), and Xylomyidae, while larvae of timber flies (Pantophthalmidae ) are saproxylic, burrowing in living wood (James, 1981; Rozkošný, 1982; Pujol-Luz & Pujol-Luz, 2014a). Xylomyidae is a small group of flies with predacious or saprophagous larvae living under tree bark (James, 1981). Pantophthalmidae, the group of timber flies, including one of the largest extant representatives of Diptera, with larvae burrowing in living wood (Rapp, 2007, 2011).

Representatives of Stratiomyomorpha are carrying vital ecosystem functions in their respective habitats: (1) the larvae often act as important saprophages, involved in the cycling of organic matter and (2) adults are important pollinators (Hauser, Woodley & Fachin, 2017).

One species of soldier flies, namely Hermetia illucens (Linnæus, 1758), with its fast-growing scavenger-type larvae, is considered as an essential source of protein for feeding cattle in industrial agriculture or for the production of human food (Hauser, Woodley & Fachin, 2017; Lessard, Yeates & Woodley, 2019). Many merolimnic species of Stratiomyomorpha, that is, those with aquatic larvae, are important algal mat grazers, involved in carbon cycling (Mángano, Buatois & Claps, 1996).

Ichnofossils attributed to larvae of Stratiomyomorpha are quite common in the fossil record (Mángano, Buatois & Claps, 1996; Pickerill, Han & Jiang, 1998), while body fossils of this group have been scarce (Evenhuis, 1994). So far, only five deposits are yielding any of them:Whalley & Jarzembowski (1985) reported four stratiomyomorphan larvae, differentiated into two morphotypes, from the Early Cretaceous Montsech (Lerida, Spain, 125.45 to 122.46 Ma) lithographic limestone of Spain.

Two morphotypes of larvae from Myanmar amber (~100 MYA) were reported by Liu, Hakim & Huang (2020).

Kühbander & Schleich (1994) reported a stratiomyomorphan larva, interpreted as a larva of the group Odontomyia, from the Miocene Randecker Maar in Germany (~17 MYA). Numerous additional specimens were recorded later from the same deposit (Rasser et al., 2013).

Karl & Bellstedt (1989) reported a single body fossil of a larva of the group Stratiomyidae from the Holocene of Eastern Germany (>1 MYA).

Sixteen fossil larvae of Stratiomyidae from the late Eocene of the Isle of Wight (129.4 ± 1.5 MYA) are present in the collection of the Natural History Museum London (UK). They can be interpreted as aquatic forms of the group Stratiomyini and have been tentatively suggested to be representatives of the species Odontomyia brodiei (Cockerell Theodore, 1915) which is known from fossils of adults from the same deposit.

Larval forms are crucial for the success and diversification of any ingroup of Holometabola, due to the ecological niche separation of the life stages (Grimaldi & Engel, 2005). This applies to the super-diverse lineages of beetles (Coleoptera), wasps (Hymenoptera), butterflies (Lepidoptera) and flies (Diptera), but also the less species-rich groups. Lack of fossil larvae of Stratiomyomorpha is hampering progress in our understanding of the evolution of the group. Here we report new records of larvae of Stratiomyomorpha based on new fossil specimens. We furthermore discuss the ecological roles of the extinct larval forms based on morphometric comparison of modern and fossil forms.

Materials and Methods

Material

Twenty-three specimens of fossil larvae are in the focus of this study. Twenty of them are preserved in amber, and three are preserved as compression fossils. These larvae are representing six morphotypes: two from Myanmar amber, one from Baltic amber, one from Dominican amber, one from the Činžat shale of Slovenia, and the last one from the Messel lake deposits. Most of the specimens in amber originated from Myanmar (“Burmese amber”), and most represent a single morphotype (“morphotype 1”). Working with Burmese amber requires special ethical consideration; for details, see the ongoing discussion (Haug et al., 2020a). All these specimens were purchased on ebay.com from different sellers and are now deposited at the collection of the Palaeo-Evo-Devo Research Group, Ludwig-Maximilians-Universität, Munich, Germany (PED) (PED-0152, PED-0349, PED-0362, PED-0031, PED-0041, PED-0243, PED-0113, PED-0025). Specimens were purchased between the 20th of March and 1st of September 2019 from two ebay.com users. Details on the purchase of the every individual amber piece (some containing multiple studied specimens) are available as ebay.com screenshots in the Supplemental Material.

A second morphotype (morphotype 2) also preserved in amber from Myanmar is represented by five larval specimens, preserved in a single piece of amber (accession number NHMLA-LACM ENT 366281). This specimen is deposited in the collection of the Los Angeles County Museum of Natural History, Los Angeles, CA, USA (LACM).

Specimen PED-0462, representing morphotype 3, was commercially acquired by Y.W. and originated from the Dominican Republic. It is now deposited in the collection of the Palaeo-Evo-Devo Research Group, Ludwig-Maximilians-Universität, Munich, Germany (PED).

Specimen PED-0463, representing morphotype 4, was collected by R.G. at the locality Činžat, situated in the Ribnica-Selnica Graben, northern Slovenia. The specimen is now deposited at the collection of the Palaeo-Evo-Devo Research Group, Ludwig-Maximilians-Universität, Munich, Germany (PED).

Specimen PED-0464, representing morphotype 5, was obtained commercially from Mr. Jonas Damzen (http://www.amberinclusions.eu) and stemmed from Yantarnyj, Kaliningrad district (formerly Palmnicken, Königsberg). It is now deposited in the collection of the Palaeo-Evo-Devo Research Group, Ludwig-Maximilians-Universität, Munich, Germany (PED).

Finally, two compression fossils originated from the Messel pit fossil site in Germany, representing morphotype 6, are deposited under coll-no. SF-MeI 4666 in the collection of the Forschungsinstitut und Naturmuseum Senckenberg (SF), Frankfurt am Main, Germany.

For comparative purposes, we used extant larval representatives of Stratiomyidae from the collection of the Zoological State Collection, Munich (Zoologische Staatssammlung München, ZSM), in particular larvae of: Pachygaster atra (Panzer & Wolfgang, 1798), Oxycera nigricornis Olivier, 1811, as well as Odontomyia sp. The latter is deposited in the collection of the Palaeo-Evo-Devo Research Group, Ludwig-Maximilians-Universität, Munich, Germany (PED-0465). For a full list of material please see Table 1.

Table 1 Material used in the paper.

Please note that ZSM does not provide numbers for most of their extant insect material, including specimens used in this article.

Taxon	ID number	Larvae	Syninclusions	Deposited at	Age	
Morphotype 1	PED-0152	3	NA	PED	Cretaceous, Cenomanian 100.5 ± 0.4 MYA	
Morphotype 1	PED-0349	1	NA	PED	Cretaceous, Cenomanian 100.5 ± 0.4 MYA	
Morphotype 1	PED-0362	1	Hymenoptera	PED	Cretaceous, Cenomanian 100.5 ± 0.4 MYA	
Morphotype 1	PED-0031	4	Diplopoda, beetle, 2 beetle larvae, collembola, aranea, probable scale insect, 2 mites	PED	Cretaceous, Cenomanian 100.5 ± 0.4 MYA	
Morphotype 1	PED-0041	1	NA	PED	Cretaceous, Cenomanian 100.5 ± 0.4 MYA	
Morphotype 1	PED-0243	1	NA	PED	Cretaceous, Cenomanian 100.5 ± 0.4 MYA	
Morphotype 1	PED-0113	1	NA	PED	Cretaceous, Cenomanian 100.5 ± 0.4 MYA	
Morphotype 1	PED-0025	1	NA	PED	Cretaceous, Cenomanian 100.5 ± 0.4 MYA	
Morphotype 2	LACM ENT 366281	5	NA	LACM	Cretaceous, Cenomanian 100.5 ± 0.4 MYA	
Morphotype 3	PED-0462	1	NA	PED	Miocene, 20.44–13.92 MYA	
Morphotype 4	PED-0463	1	NA	PED	Miocene, 20.44–15.97 MYA	
Morphotype 5	PED-0464	1	non-biting midge male (Diptera, Chironomidae); window-gnat (Diptera, Anisopodidae), two dark-winged fungus gnats (Diptera, Sciaridae), large spider (Araneae)	PED	Eocene, 37.8–33.9 MYA	
Morphotype 6	SF-MeI 4666	2	NA	SF	Eocene, ca. 48.2 MYA	
Pachygaster atra (Panzer, 1798)	not provided	>100	NA	ZSM	extant	
Oxycera nigricornis Olivier, 1811	not provided	>100	NA	ZSM	extant	
Odontomyia sp.	PED-0465	1	NA	PED	extant	

Terminology: The morphological terminology mostly follows Rozkošný (1982), and Sinclair (1992) for the head capsule morphology. Yet, to help non-experts, we amended some of the special morphological terms with more general terms. As Insecta is an accepted ingroup of Crustacea s.l., “crustacean” structure names are given in brackets where necessary to provide more comprehensive frame correspondence. It is important to note that many structures cannot be discerned externally in the Diptera larvae, that is, it is impossible to see any tergite boundaries in the head capsule of the post-embryonic larvae. Nevertheless, it is well possible to reconstruct the sequence of the segments in the head capsule, using the arrangement of the appendages (Baranov, Schädel & Haug, 2019).

Database use

Data on the fossil record of the group Stratiomyomorpha were downloaded from the Paleobiology Database on 9 November 2019, using the group name “Stratiomyomorpha” without any other filtering parameters.

Imaging methods

Amber specimens were imaged using a Keyence VHX-6000 Digital microscope, with ring-light type illumination and/or cross-polarized, coaxial illumination. All images were recorded as composite images to counteract the limitations of depth of the focus. Models were assembled using stitching and panorama functions to overcome the weakness of the field of view under higher magnifications. Each image detail was recorded by a stack of images of shifting focus to overcome the limitation of the depth of field (Haug, Haug & Ehrlich, 2008; Haug et al., 2011; Haug, Müller & Sombke, 2013). Fusion into sharp images and panorama stitching was performed with the built-in software, for example, in Baranov, Schädel & Haug (2019). We also employed the built-in HDR function of the digital microscope; therefore, every single frame is a composite from several images taken under different exposure times (cf. Haug, Müller & Sombke, 2013). Additionally, some specimens were imaged using a Keyence BZ-9000 fluorescence microscope with either 2×, 4×, 10× or 20× objectives. Observations were conducted at an emitted wavelength of 532 nm since it was the most compatible with the fluorescence capacities of the fossil specimens (Haug et al., 2011). Also, here we recorded stacks of images which then were digitally fused to single in-focus images using CombineZP (GNU). Compression fossils from Messel were photographed with a Leica MZ12.5 stereomicroscope with an attached Nikon D300 camera.

The cuticle fossil, specimen PED-0463 (“morphotype 4”), was additionally imaged using a scanning electron microscopy (SEM). Scanning was performed using a Carl Zeiss Leo 1430VP scanning electron microscope in the Zoologische Staatssammlung München (Germany). Scanning was performed with the beam current 80 µA; filament electric current 2,500 A; and electric potential 10–20 kV. Scanning was performed in a low vacuum (<2e−005 mbar).

Morphometry and outline analysis

The maximum dorsal head capsule length and width of some larvae were measured from the tip of the labrum to the outer edge of the head capsule. Actual measurements were done from the photos, using ImageJ, a public domain software (Schindelin et al., 2012).

As a proxy for the overall shape diversity we compared the outlines of the larvae in the dorsoventral aspect. To do so, we have analyzed the shapes, more precisely sketches of all specimens, and extant comparative specimens with Fourier Elliptical Transformation using R package Momocs (Bonhomme et al., 2014) and compared morphospace occupancy.

For the outline analysis we used black-and-white .jpg files, containing the outlines of all available fossil stratiomyomorphan larvae as well as all extant stratiomyomorphan larvae for which we were able to obtain a full-body image in the dorsoventral aspect from the literature. Only specimens with a relatively straight body were included, as any examples imaged in curled or bent position will heavily bias the morphospace. Full-body images of the larvae were obtained from numerous published sources (Schremmer, 1951, 1984; Hennig, 1952; James, 1965; McFadden, 1967; Bull, 1976; Teskey, 1976; James, 1981; Beuk, 1990; Dušek & Rozkošný, 1967; Rozkošný, 1997; Rozkošný & Kovac, 1998; Pujol-Luz & Xerez, 1999; Stubbs & Drake, 2001; Stuke, 2003; Rozkošný, 1983; Pujol-Luz, De Xerez & Viana, 2004; De Xerez & Garcia, 2008; Bucánková, Kovac & Rozkošný, 2009; Marques & De Xerez, 2009; Marshall, 2012; Pujol-Luz & Pujol-Luz, 2014a, 2014b; Pujol-Luz, Lopes & Viana, 2016; De Godoi et al., 2018), see Supplementary Table 1 for the full information. In total 69 stratiomyomorphan specimens were analyzed (see Table S1, supplementary images).

Black-and-white outlines were produced using a polygonal tool and mask functionality of the program FIJI (Schindelin et al., 2012). Jpg outlines were analyzed in R using the momocs package (Bonhomme et al., 2014), with the shapes being characterized by 36 harmonics. Source code, list of the material used for the outline production and all the underlying data are available as Supplemental Material (Supplemental Information 3). To estimate the habitat affinity of the fossil larvae, we plotted them into a single morphospace with the extant larvae. For the latter, we demarcated saproxylic, aquatic, and terrestrial habitats. Based on the position of the fossil larvae in this morphospace, we have attempted to assess their habitat affinity. All data analyses were conducted in R version 3.4.1 (2017-06-30) - “Single Candle” (R Core Team, 2014).

Data availability

All the specimens used in the paper are deposited in public collections (see Table 1). All the outline jpg images are provided in the Supplemental Materials to this paper, together with the R code used to conduct the analysis.

Geological context

The geological context of Myanmar (Cruickshank & Ko, 2003; Yu et al., 2019), Dominican (Iturralde-Vinent, 2001) and Baltic (Wichard, Gröhn & Seredszus, 2009) ambers, as well as Messel shale (Schaal, Smith & Habersetzer, 2018), has been explained in detail in various previous works.

The Locality of Činžat is much less well known to the broader audience than the three above mentioned, so we are discussing it in further detail. The studied locality Činžat is situated in the Ribnica-Selnica graben (Jelen & Rifelj, 2002) filled with sediments once deposited in the Central Paratethys sea (Rögl, 1999), within the westernmost parts of the Styrian Basin, approximately 15 km west of Maribor. Here, strata of the Ivnik Beds (Mioč, 1972) are exposed in a belt from Maribor, on the northern slopes of the igneous Pohorje pluton, towards the town of Radlje.

Fossil bearing micaceous laminated siltstones cover older pre-Cenozoic rocks and sequences of loosely bound conglomerates, alternating with sandstones and siltstones of the Ivnik Beds. A late Burdigalian age (Miocene) coinciding with the “Karpatian” stage of the regional scale was identified based on a benthic formainiferan association and nannoplankton sampling (Gašparič & Hyžný, 2015).

The fossil fauna includes decapod crustaceans, bivalves, gastropods, and echinoids, which are randomly distributed within the siltstone layers of the Činžat section, although individual layers and variations in lithology are more likely to have macrofaunal fossil remains. Interbedded layers of sandstones and conglomerates contain no macrofossils. The faunal association suggests low energy deep-water depositional environment with epibathyal water depth exceeding 125 m (Gašparič & Hyžný, 2015).

Results

In total we can distinguish six different morphotypes among the studied fossil larvae.

Description, general notes: to provide the necessary background, we first give a generalized description of the characteristics of larvae of Stratiomyidae (and to some extent Stratiomyomorpha) segment by segment.

General shared appearance

Habitus. Small to medium-sized larva with slightly dorsoventrally flattened, spindle-shaped body. The body is fully covered with oval pellets, supposedly of calcium carbonate composition (although it is impossible to ascertain this aspect for the fossil forms) (Figs. 1A, 1B, 2A–2C, 3A-3C and 4A).

Figure 1 Morphology of larva of the group Stratiomyidae, exemplified by a larva of Pachygaster atra.

(A) Ventral view, marked; (B) dorsal view, marked. Abbreviations: a1–a7, abdomen units one through seven; ap, anal setae; asl, anal slit, as anterior spiracle; D1–D3, dorsal setae 1–3; DL, dorsolateral setae; ep, eye prominence; hc, head capsule; L, lateral setae (of abdomen unit); L1–L2, lateral setae (of trunk end); mp, maxillary palp; mt, metathorax; pt, prothorax; sa, subapical setae; v1–v4, ventral setae one – through four (of the trunk end); VL, ventrolateral setae (of the abdomen units 1–7).

Figure 2 Larva of morphotype 1, specimen PED-0031-2.

(A) Ventro-lateral view; (B) Ventro-lateral view, marked; (C) Dorso-lateral view. Abbreviations: hc, head capsule; as, anterior spiracle; pt, prothorax; mt, metathorax; a2–a6, posterior trunk units 2–6; te, trunk end; ps, posterior spiracle.

Figure 3 Larva head of morphotype 1, specimen PED-0031-2.

(A) Lateral view; (B) Lateral view, marked; (C) Lateral view, linedrawing. Abbreviations: hs, head soft tissues; hc, head capsule; lb, labrum; la, labium; mk, mandibular-maxillar complex; ta, tentorial arm; mr, metacephalic rode; ct, cut off through the part of the head capsule; pt, prothorax; cc, calcium carbonate pallet.

Figure 4 Larval head of morphotype 1, specimen PED-0152-2.

(A) Dorsal view, habitus; (B) Ventral view, head; (C) Ventral view, head-marked. Abbreviations: mp, maxillary palp; bm, base of mandibular-maxillar complex (“grinder”); lb, labrum.

Body length from 2 mm to slightly less than 20 mm. The body differentiated into presumably 20 segments, ocular segment plus 19 post-ocular segments (Figs. 1A, 1B, 2A–2C and 4A–4C). Anterior segments are forming a distinct head capsule.

The head capsule sclerotized anteriorly, posterior part (one third to one half) reduced to several longitudinal structures, retracted into the anterior trunk (prothorax). The head capsule is formed by an ocular segment plus five post-ocular segments.

Ocular segment recognizable by its appendage derivative, clypeo-labral complex. The clypeus (clypeal sclerite) is longer than it is wide. Labrum is roughly triangular, much longer than wide, strongly sclerotized (Figs. 2A–2C, 3B and 3C). The segment with small stemmata (“eye prominences”), anteromedially.

Post-ocular segment 1, with a pair of antennae [antennulae in generalized terminology]. The antenna of Stratiomyomorpha larvae stout, comprising two elements, sitting in dorsoanterior position, or more towards the center of the dorsal surface of the head capsule. In many fossil specimens not preserved or not visible (Figs. 4B and 4C).

Post-ocular segment 2 (intercalary segment) without externally recognizable structures, not identifiable in the post-embryonic development of most larvae of Diptera. It might be argued that discussion of such seemingly absent structure in the description is unnecessary, or mixing conjectures with observed structures. We will say, on the contrary: there is the knowledge and hence expectation of the presence of this segment based on prior experience. Yet, we do not see it. In broader comparison, this is, in fact, informative and needs to be included in the taxa description.

Post-ocular segments 3 and 4 were recognizable by their appendages: mandibles and maxillae [maxillula in generalized terminology]. Mandibles and maxillae form a single compound, the mandibular-maxillary complex (autapomorphy of Stratyomyomorpha), comprising elements largely indistinguishable, apart from the distal parts of the maxillae (maxillary palp). Maxillary palps quite stout, but prominent (Schremmer, 1951). The proximal portion of the mandibular-maxillary complex, fully sclerotized; with strong, multi-branched setae on its dorsodistal surface, as well as laterally. Distal part or palp conical, with two elements (palpomeres). The apical part of the mandibular-maxillary complex bears an arrangement of the setae (“brush”) of varying complexity (autapomorphy of Stratyomyomorpha). Basal part bearing a ventral "grinder", which is heavily sclerotized (Figs. 3A–3C, 4B and 4C). On the ventral side, the mandibular-maxillary complex forms ventral plates, occupying the ventral side of the head capsule (Sinclair, 1992).

Trunk (thorax + abdomen) with eleven visible units, interpreted as three thorax segments, seven abdominal segments and a trunk end (abdomen unit 8). Cuticle covered with round deposits of calcium carbonate pellets, forming a honeycomb-like pattern (autapomorphy of Stratyomyomorpha). Remark: It is difficult to ascertain that the cuticle of the fossils is indeed covered in calcium carbonate pellets. It cannot be excluded that such cuticle scales are simple chitin as in larvae of the Ephydridae or Oestridae (Marshall, 2012).

Trunk units are without any parapods, creeping welts or protuberances. Different arrangements of spiracles possible: (1) Trunk bears nine pairs of spiracles (openings of the tracheal system): one pair of spiracles on thr prothorax, and eight pairs on the posterior trunk (abdomen). This type of tracheal system is called peripneustic (Hennig, 1952). (2) Most spiracles reduced, amphipneustic (spiracles present on prothorax and trunk end) or (3) methapneustic (spiracles on trunk end only) (McFadden, 1967) (Figs. 2A–2C; Figs. S4A–S4D).

The anterior trunk or thorax has three segments, pro-, meso- and methathorax. Armament represented by the calcium carbonate pellets and large rhombic sclerites on the sternites, occasionally with some modified, spike-like setae. Prothorax with 2–3 pairs of anterodorsal setae.

Mesothorax and metathorax have numerous dorsal and ventral setae as well as multiple pellets of calcium carbonate (Figs. 1A and 2B). All three units (= segments in this case) of the anterior trunk (thorax) are having very uniform setation. Each of the thoracic segments bears three pairs of dorsal setae (D1–D3), one pair of dorsolateral setae, and one pair of ventrolateral setae. Additionally, each thoracic segment bears two pairs of ventral setae (Figs. 1A and 2B). The inner pair of ventral setae simple; the outer pair contains several branched setae. The latter also is known as “thoracic leg group” setae.

Posterior trunk (abdomen) units 1–7 with setae arranged in a uniform pattern. This pattern includes three pairs of dorsal setae, in addition to a single pair of dorsolateral and a pair of ventrolateral setae on each of the abdominal units. Each of these units also bears one or two pairs of lateral setae. These lateral setae can be quite prominent. Additionally, three (sometimes four) pairs of ventral setae arranged in a transverse row on the sternites of abdominal units 1–7. The trunk end (abdomen unit 8) bears two pairs of lateral setae, which are often quite long. Additionally, the trunk end bears one pair of subapical setae, and one pair of apical setae. The setae of both groups are usually relatively short. Dorsal setae are present but rarely prominent on the trunk end. A large anal cleft (anus) present on the ventro-terminal part of the trunk end. Around this cleft, the ventral setae arranged in five pairs, situated along and behind the anal cleft (Fig. 1B).

Summary of main results

In total, we can distinguish six different morphotypes among the studied fossil larvae.

Morphotype 1 (Stratiomyomorpha)

Material examined: 13 specimens (PED-0025, PED-0031_1, PED-0031_2, PED-0031_3, PED-0031_4, PED-0041, PED-0113, PED-0152_1, PED-0152_2, PED-0152_3, PED-0243, PED-0349, PED-0362) in 8 amber pieces (see Table 1; Fig. S1). Most of the measurements were performed on the two best preserved specimens PED-0031_1 and PED-0041 (Figs. 2A–2C, 3A–3C and 4A–4C; Figs. S1–S7). Syninclusions: see Table 1.

Description:

Habitus. Medium-sized larva with slightly dorsoventrally flattened, spindle-shaped body, fully covered with oval pellets or scales (Figs. 2A–2C and 4A).

Body length 2.3–1.1 mm (n = 9). The body differentiated into presumably 20 segments, ocular segment plus 19 post-ocular segments (Figs. 2A–2C and 4A). Anterior segments forming a distinct head capsule.

The head capsule sclerotized anteriorly, and the posterior part is reduced to several longitudinal structures (unpaired metacephalic rod, paired tentorial arms) retracted into the anterior trunk (prothorax and mesothorax). Dimensions of head capsule (including metacephalic rod and the tentorial arms protruding far back into the prothorax) (Figs. 4A–4C).

Ocular segment recognizable by its appendage derivative, clypeo-labrum complex. Clypeus (clypeal sclerite) longer than wide. Labrum is roughly triangular, much longer than wide, strongly sclerotized (Figs. 3A–3C, 4B and 4C). Segment with small apparent stemmata, posterolaterally.

Post-ocular segment 1 recognizable by its pair of appendages, antenna [antennula]. Antenna prominent, robust 25 µm long (n = 1) (Figs. 4B and 4C).

Post-ocular segment 2 (intercalary segment) without externally recognizable structures.

Post-ocular segments 3 & 4 are recognizable by pairs of appendages, mandibular-maxillary complex (Figs. 2A–2C and 3A–3C). The proximal part fully sclerotized with strong, multi-branched setae on the dorsodistal surface, as well as laterally. Distal part, palp, conical, with two elements (palpomeres). Basal part of the complex bearing a large molar “grinder”, which is occupying the almost entire ventral side of the head capsule and heavily sclerotized (Figs. 2A–2C, 3B and 3C).

Post-ocular segment 5 recognizable by its appendages, forming the labium, represented by a fleshy lobe.

Trunk (thorax+abdomen) with 11 visible units, interpreted as three thorax segments, seven abdomen segments and a trunk end (abdomen unit 8). The trunk is yellowish-brown, except for the very first unit which is light-yellow. The cuticle is covered with oval pellets or scales. Units of the posterior trunk do however bear complex armament on dorsal and ventral sclerites (tergites and sternites; Figs. 1A–1C).

Anterior trunk, thorax with three segments, pro-, meso- and metathorax.

The prothorax 450–770 µm long (n = 2), without protrusions. Oval pellets or scales represent armament, and large rhombic sclerite on the sternite, with two rows of small, flat spikes arranged anteriorly on sternite. Distinct spiracle (anterior spiracle) on shallow depression at the posterolateral part of the prothorax (Figs. 1A–1C).

Mesothorax yellowish-brown, 360–540 µm long (n = 2). With two rows of triangular, flat spines on the anterior edge of sternite. Numerous oval pellets or scales.

Metathorax yellowish-brown, 400–660 µm long (n = 2), bears two rows of triangular, flat spines on the anterior edge of sternite, as well as numerous oval pellets or scales.

Posterior trunk, abdomen with eight distinct units. Anterior seven representing proper segments.

Abdomen unit 1 rectangular in dorsoventral plain, 440–760 µm long (n = 2). Bearing numerous oval pellets or scales, as well as two rows of the small triangular spikes on the anterior edge of the sternite. Posterior edge of dorsal sclerite, tergite, with a row of 12 robust, dorsoventrally triangular spines.

Abdomen units 2–7 rectangular (370–920 µm long). Bearing numerous oval pellets or scales. Posterior edge of dorsal sclerites, tergites, each with a row of robust triangular spines, 10–12 such spines on abdomen units 2–6, 7 on abdomen unit 7.

Trunk end (abdomen unit 8, undifferentiated abdomen segments 8–11?) roughly trapezoid in the dorsoventral view, 620–750 µm long (n = 2). With three pairs of small lateral setae, two pairs of strong black setae on two mounds at the middle of the tergite; two pairs of strong needle-like setae on two smaller mounds at the distal edge of on dorsal tergite (syn-tergite?). Tergite also bears posterior spiracles in a transversal cleft, ventrally. Large, transversal anal cleft, surrounded by an elevated oval sclerotized area, of a markedly darker color than the rest of the cuticle visible at the trunk end.

Morphotype 2 (Stratiomyomorpha: Stratiomyidae)

Material examined: LACM ENT 366281 (five specimens in a single piece). Most measurements are based on a single specimen, well preserved and visible in dorsal aspect (Figs. 5A, 5B, 6A–6D; Figs. S8A and S8B).

Figure 5 Larva of morphotype 2, specimen LACMALACM ENT 366281-1.

(A) Dorsal view, habitus; (B) Dorsal view, habitus-marked. Abbreviations: hc, head capsule; as, anterior spiracle; ms, mesothorax; mt, metathorax; a1–a7, posterior trunk units 1–7, te, trunk’s end; ps, posterior spiracle.

Figure 6 Larval head of morphotype 2, specimens LACM ENT 366281-1 (A and B) and LACM ENT 366281-2 (C and D).

(A) Dorsal view, head, LACM ENT 366281-1; (B) Dorsal view, head-marked, LACM ENT 366281-1; (C) Lateral view, head, LACM ENT 366281-2; (D) Lateral view, head-marked, LACM ENT 366281-2. Abbreviations: hs, head soft tissues; hc, head capsule; lb, labrum; mk, mandibular-maxillar complex; ey, eyes; as, anterior spiracle.

Syninclusions: NA

Description:

Habitus. Medium-sized larva with somewhat dorsoventrally flattened, spindle-shaped body, covered with oval pellets of the calcium carbonate (Figs. 5A and 5B).

Body covered by the white film, precluding observation of many fine details. Length 3.3–3.7 mm (n = 3). Body differentiated into presumably 20 segments, ocular segment plus 19 post-ocular segments (Figs. 5A and 5B; Figs. S8A and S8B). Anterior segments are forming a distinct head capsule.

The head capsule sclerotized anteriorly, posterior part reduced to several longitudinal structures (unpaired metacephalic rod, paired tentorial arms), retracted into prothorax. Dimensions of head capsule: 480 µm long, 340 µm wide (n = 1).

The surface of the head capsule is covered with pellets of calcium carbonate (Figs. 6A–6D).

Ocular segment recognizable by its appendage derivative, clypeo-labral complex. Clypeus (clypeal sclerite) fuzed with the frontal sclerite. Labrum roughly beak-like (100 µm long, 70 µm wide), much longer than wide, strongly sclerotized (Figs. 6A–6D). A segment with small apparent stemmata, anterolaterally.

Post-ocular segment 1: not externally recognizable, possible structures (antennae) not apparent.

Post-ocular segment 2 (intercalary segment) without externally recognizable structures.

Post-ocular segments 3 & 4 are recognizable by their pairs of appendages forming a mandibular-maxillary complex (Figs. 6A–6D). Distal lobe brown in color, distal ends chisel-like. Palp (distal part) not visible on any of the specimens available (Figs. 5A–5D).

Post-ocular segment 5 is not recognizable, its appendages, presumably forming the labium, not visible in any of the specimens available (Figs. 6A–6D).

Trunk (thorax+abdomen) with 11 visible units, interpreted as three thorax segments, seven abdomen segments and trunk end (abdomen unit 8). Trunk is yellowish-brown, except for the very first unit which is light-yellow. Cuticle is covered with round deposits of calcium carbonate pellets. Trunk dorsoventrally flattened, spindle-shaped, total length 1.9–2.7 (n = 3) mm long; densely covered with oval pellets or scales (Figs. 5A and 5B; Figs. S8A and S8B).

Anterior trunk, thorax with three segments, pro-, meso- and metathorax.

The prothorax is ring-like, 240 µm long, 630 µm wide (n = 1), with ventral excision at place of head capsule insertion. Small spiracles on posterolateral surface. Prothorax bears no protrusions. Oval pellets or scales represent the armament. The anterior spiracle sits on a conical protrusion, ca. 35 µm long, spiracle itself with a single longitudinal opening (Figs. 5A and 5B; Figs. S8A and S8B).

Mesothorax is 110 µm long, 780 µm wide (n = 1), ring-shaped, with no visible protrusion, bearing numerous oval pellets or scales.

Metathorax is 180 µm long, 820 µm wide (n = 1), ring-shaped, with one pair of the long, wavy setae.

Posterior trunk, abdomen with eight distinct units. Anterior seven representing true segments.

Abdomen units 1–7 are wider than long (200–260 µm long; 900–1000 µm wide). All units are bearing several wavy lateral setae; unit 7 additionally bears two lateral wavy setae.

Trunk end (abdomen unit 8, undifferentiated abdomen segments 8–11?) roughly square shaped in dorsal or ventral view (502 µm long, 525 µm wide); with two pairs of the large, wavy setae. Anal cleft is sitting on large elevated mounds posteriorly on tergite (Figs. 5A and 5B; Figs. S8A and S8B).

Morphotype 3 (Stratiomyomorpha: Stratiomyidae)

Material examined: Piece of Dominican amber with a single fossil larva from the PED collection (collection number PED-0001; Figs. 7A, 7B, 8A–8D and 9; Figs. S9A, S9B, S10A and S10B). The larva is well preserved, anterior trunk obscured ventrally by a large air bubble. Head capsule details inaccessible.

Figure 7 Morphotype 3, habitus, ventral, larva PED-0462.

(A) Habitus, ventral view; (B) Habitus, ventral view, markes. Abbreviations: hc, head capsule; pt, prothorax; ms, mesothorax; mt, metathorax; a1–a7, abdominal units 1–7; te, trunk’s end.

Figure 8 (A and B) Fossil Pachygastrinae, larva of morphotype 3, PED-0462 and (C and D), extant Pachygaster atra, head.

(A and B) Ventral view of the headcapsule unmarked and marked; (C and D) Pachygaster atra, Ventral view of the headcapsule unmarked and marked; Abbreviations: an, antenne; as, anal setae; pt, prothorax; ey, eyes; lb, labrum; mp, maxilar palp; mb, base of mandibular-maxillar complex (“grinder”); v1–3, ventral setae 1–3; la, labium.

Figure 9 Speculative reconstruction of the habitus and habitat of the fossil larva of the group Pachygastrinae, morphotype 3.

Onychophora Tertiapatus sp. stalking at the background. Artwork by Christian McCall, reproduced with permission.

Description:

Habitus. Medium-sized larva with dorsoventrally flattened body, and triangular posterior end in the dorsoventral plain) (Figs. 7A, 7B, 8A, 8B and 9). The body armored with the oval pellets or scales. Total length 9.5 mm. Body differentiated into presumably 20 segments, ocular segment plus 19 post-ocular segments (Figs. 7A and 7B). Anterior segments forming a distinct head capsule.

The head capsule partially sclerotized, longer than wide, posterior part of the head capsule is retracted into the trunk. Dimensions of head capsule: 720 µm long, 550 µm. The surface of the head capsule is covered with small cuticular scales with oval pellets or scales (Figs. 7A and 7B).

Ocular segment recognizable by its appendage derivative, clypeo-labral complex. With two pairs of setae, two labral setae and two frontoclypeal setae. Clypeus continuous with labrum, clypeus narrow, labrum expanding distally (Figs. 8A and 8C). Segment with pair of apparent stemmata (larval eyes).

Post-ocular segment 1 is recognizable by its pair of appendages, antennae [antennula], inserting ventrolaterally at the anterior end of the head capsule (Fig. 8B). Antenna short, consists of two elements.

Post-ocular segment 2 (intercalary segment) without externally recognizable structures.

Post-ocular segments 3 & 4 are recognizable by pairs of appendages, mandibular-maxillary complex. Proximal part is heavily sclerotized, with basal plates. Main part of the lobe hook-shaped, continuous, with appendages of the following post-ocular segment, integrated into the mandibular-maxillary complex. The inner surface forms longitudinal striated “molar” area (Figs. 8A and 8B). Distal lobe is fleshy, with numerous maxillary setae (Figs. 8A and 8B).

Post-ocular segment 5 recognizable by its appendages, forming the labium. Labium bearing 3 pairs of setae (2 ventral setae and 4 ventrolateral). Proximal part of a labium forms a funnel connected to the oral cavity. Labium distally with two projections, probably palps. Labium is highly modified, connected to cibarial (pharyngeal) skeleton of the head capsule (Figs. 8A and 8B).

Trunk (thorax+abdomen) has 11 visible units, interpreted as three thorax segments, seven abdominal units and a trunk end (abdominal segment 8) (Figs. 7A, 7B and 8A–8D; Figs. S9A, S9B, S10A and S10B). The trunk is spindle-shaped in a dorsoventral plain, parallel sided in the middle region, triangular at the hind-end. All units are bearing oval pellets or scales and long setae.

Anterior trunk (thorax) has three segments: pro-, meso- and metathorax. Thoracic “leg” setae groups are seemingly with two setae in each group (Figs. 7A, 7B and 9; Figs. S9A, S9B, S10A and S10B).

Prothorax is 760 µm long. Prothorax with numerous setae: four antero-dorsal, six dorsal, two dorsolateral, four lateral, two ventrolateral and six ventral (Figs. 7A and 7B; Figs. S9A, S9B, S10A and S10B). Prothorax bears a pair of spiracles.

Mesothorax is 800 µm long, with numerous setae: six dorsal, two dorsolateral, four lateral, two ventrolateral and six ventrals.

Metathorax is 500 µm long, with numerous setae: six dorsal, two dorsolateral,

four lateral, two ventrolateral and six ventral setae (Figs. 6A and 6B; Figs. S9A, S9B, S10A and S10B).

Posterior trunk (abdomen) has eight apparent units flattened dorsoventrally, mostly oval in the dorsal plain, with triangular posterior hind-end (Figs. 7A, 7B and 9). Abdomen units 1–7 are with numerous setae: six dorsal setae, two dorsolateral setae, four lateral setae, two ventrolateral setae four ventral, on each segment (Figs. 7A and 7B; Figs. S9A, S9B, S10A and S10B).

Trunk end (abdomen unit 8, undifferentiated abdomen segments 8–11?) is triangular in general shape, dorsoventrally,it has well visible anus on the ventroterminal part. The trunk end carries numerous setae: ventral setae pairs v1–v4, two pairs of anal setae and eight dorsolateral setae. The terminal end is elongated into the two rod-shaped protrusions, each carrying anal setae. No cuticular “teeth” are present along the anal opening (Figs. 7A and 7B).

Morphotype 4 (Stratiomyomorpha: Stratiomyidae)

Material examined: small slab of the Činžat shale with a cuticular fossil of a larva. Specimen split in half along the medio-lateral surface of the sternites, so that tergites of the posterior trunk (units 5–8) are folded upon the tergites of the more anterior ones (1–4). Coloration of specimen very well preserved (Figs. 10A, 10B and 11A–11D; Figs. S12A, S12B and S13A–S13D).

Figure 10 Pachygastrinae, larva, morphotype 4, PED-0463.

(A) Habitus, ventral view; (B) habitus, ventral view, marked. Abbreviations: hc, head capsule; ey, eyes; as, anterior spiracle; pt, prothorax; ms, mesothorax; mt, metathorax; a1–a7, posterior trunk units 1–7; te, trunk’s end; ps, posterior spiracle; fc, folded cuticle.

Figure 11 Pachygastrinae, larva, morphotype 4, PED-0463, head ventrally.

(A) Head and prothorax, ventral view; (B) head and prothorax, ventral view, marked; (C) head, ventral view; (D) head, ventral view, marked. Abbreviations: an, antenna; hc, head capsule; ey, eyes; as, anterior spiracle; lb, labrum; la, labium; mk, mandibular-maxillar complex.

Description:

Habitus. Medium-sized larva with dorsoventrally flattened body and rounded posterior end. Body armored with oval pellets or scales. Total length 6.4 mm. Body differentiated into presumably 20 segments, ocular segment plus 19 post-ocular segments (Figs. 10A and 10B). Anterior segments forming a distinct head capsule.

Anterior body visible in ventral perspective only, of the posterior part of the body similarly, only the dorsal region can be seen. The body with distinct sclerites ventrally on the anterior trunk, as well as dorsally on posterior trunk, bearing distinctly “leopard” pattern of coloration. This pattern consists from dark-grey and brownish-yellow spots of irregular shape (Figs. 10A and 10B; Figs. S12A and S12B).

The head capsule is sclerotized, much longer than wide, posterior part of the head capsule is retracted into the trunk. Dimensions of head capsule: 1,000 µm long, 250 µm wide (Figs. 11A–11D; Fig. S13B).

Ocular segment is recognizable by its appendage derivative: clypeo-labral complex. Clypeus continuous with labrum, narrow and blade-shaped (Figs. 11A–11D). The ocular segment has a pair of apparent hemispherical stemmata (larval eyes), at about its mid-length, dorsolaterally. The segment surface bears multiple small setae.

Post-ocular segment 1 is recognizable by its pair of appendages, antennae [antennula]. Antenna inserted dorsolaterally at the distal end of the head capsule (Figs. 11A–11D; Figs. S13B). Antenna short, with two elements.

Post-ocular segment 2 (intercalary segment) without externally recognizable structures.

Post-ocular segments 3 & 4 are recognizable by pairs of appendages, forming the mandibular-maxillary complex. The complex with a proximal lobe, heavily sclerotized, with basal plates. The main part of the basal plate is a lobe, hook-shaped, continuous with appendages of the following post-ocular segment. The inner surface forms longitudinal striated “molar” area (Figs. 11A–11D; Fig. S13B). Distally, the complex has fleshy lobe with numerous setae (Fig. 11A).

Post-ocular segment 5 is recognizable by its pair of appendages, forming the labium. Labium bears three pairs of setae (two ventral setae and four ventrolateral), on the ventral and lateral surface respectively. Proximal part of the labium forms a three-pronged structure, adjacent to the oral cavity (Figs. 10A–10D; Figs. S13B).

Trunk (thorax+abdomen) has 11 visible units, interpreted as three thorax segments, seven abdomen segments and a trunk end (abdomen unit 8). Trunk bears two pairs of spiracles (openings of the tracheal system) (Figs. 10A and 10B; Figs. S12A, S12B, S13A and S13B).

Anterior trunk (thorax) consists of three segments, pro-, meso- and metathorax. Tergites and sternites are sclerotized, bearing oval pellets or scales.

Prothorax is 300 µm long. Prothorax bears a pair of large spiracles (100 µm in diameter at the opening). Prothorax has several small setae on the dorsal surface (Figs. 10A and 10B; Figs. S12A, S12B, S13A and S13B).

Mesothorax is 300 µm long, ring-shaped, bearing no spiracles, with lighter area in the center of the sternite (probably due to the sediment filling the depressions of the fossil).

Metathorax is 250 µm long, ring-shaped, with the lighter area in the center of the sternite (probably due to the sediment filling the depressions of the fossil).

Posterior trunk (abdomen) with eight distinct units. Anterior seven units representing proper segments. Posterior trunk mostly oval with rounded posterior hind-end (Figs. 10A and 10B; Figs. S12A, S12B, S13A and S13B).

Abdomen units 1–7 are 320–610 µm long. Cuticle is split along the lateral side, medio-laterally; therefore units 5–7 (and the trunk end) are folded over the ventral parts of the units 1–4. This damage reveals the inner dorsal surface of the abdomen units 5–7 (and the trunk end) for direct observation.

Trunk end (abdomen unit 8, undifferentiated abdomen segments 8–11?) semicircular in general shape (dorsoventral view). The trunk end has an anus on ventroterminal part. No cuticular “teeth” are present along anal opening (Figs. 10A and 10B; Fig. S13C).

Morphotype 5 (Stratiomyomorpha: Stratiomyidae: Stratiomyinae)

Material examined: a single fossil larva in a piece of Baltic amber from the PED collection (collection number PED-0464). The larva is poorly preserved, covered with air bubbles and cracks in amber; only rear end of the trunk visible well enough to provide any distinguishable features (Figs. 12A and 12B). The piece of amber contains several syninclusions: non-biting midge male (Diptera, Chironomidae), window-gnat (Diptera, Anisopodidae), two dark-winged fungus gnats (Diptera, Sciaridae), large spider (Araneae).

Figure 12 Fossil larva of Stratiomyinae, morphotype 5 (PED-0464).

(A) Habitus; (B) Close-up photo of coronet of the hydrofuge setae.

Description:

Habitus. Medium-sized larva with spindle-shaped body in dorsoventral view, end of the trunk has prominent coronet of large setae. The body mostly obscured by cracks and bubbles in the amber; only rear end is clearly visible. Total length is 4.3 mm. Body differentiated into presumably 20 segments, ocular segment plus 19 post-ocular segments (Figs. 12A and 12B).

Trunk (thorax+abdomen) spindle-shaped, parallel sided, rounded at the hind-end in dorsoventral view. Anterior part of the trunk entirely obscured by cracks. Subdivision of the units unclear. Posterior trunk has densely arranged strong setae. The trunk end (undifferentiated abdominal segments 8–11?) rounded in general shape, carries strong coronet formed by 19 unbranched setae (Figs. 12A and 12B). Additionally, the trunk end bears a pair of large spiracles, surrounded by the coronet of setae and upper and lower sclerotized “lips”.

Morphotype 6 (possibly Stratiomyomorpha: Stratiomyidae)

Material examined: two fossil larvae on one slab from the Grube Messel, stored in the S.F. collection (collection number SF-MeI 4666; Figs. 13A and 13C). The fossils are originating from the Messel Formation. Specimens were collected in the year 1994 in grid square E8, 0.9 m to 1.1 m below local stratigraphic marker horizon alpha. The larvae are poorly preserved, only traces of the head capsules and the rest of the bodies can be seen; no traces of any setae are preserved; nevertheless, both specimens show a well-preserved coloration pattern of the tergites.

Figure 13 Fossil stratiomyinae larvae, morphotype 6 (SF-MeI 4666).

(A) Compression fossil, habitus; (B) compression fossil, marked; (C) Messel shale with Stratiomyiinae larvae, SF-MeI 4666, overview. Abbreviations: hc, head capsule; phc, pharyngeal grinding mill; pt, prothorax; mt, methathorax; a1–a6, abdominal units; te, trunk’s end.

Description:

Habitus. Medium-sized larva with spindle-shaped body. Accessible only in dorsal aspect.

Body length is 3.0–3.5 mm (n = 2). Body differentiated into presumably 20 segments, ocular segment plus 19 post-ocular segments (Figs. 13A–13C). Anterior segments forming a distinct head capsule.

Head capsule is partially sclerotized, longer than wide, posterior part of the head capsule retracted into the anterior trunk (prothorax). Head capsule visible only in vague outlines, with several longitudinal structures (unpaired—metacephalic rod, paired—tentorial arms). These structures are heavily sclerotized. Posterior part of the head capsule is more heavily sclerotized (Figs. 13A–13C). Width of head capsule is ca. 270 µm. Other units of the body are difficult to measure due to the poorly visible borders between the segments.

Anterior segments not apparent, without prominent structures.

Post-ocular segment 5 is recognizable by an internally located pharyngeal grinding mill (visible in both fossil specimens; Fig. 13C).

Trunk (thorax+abdomen) spindle-shaped, parallel-sided, rounded at the hind-end. With eleven units: three thorax segments, seven abdomen segments plus trunk end. Units of the trunk are much wider than long. No setae preserved. No traces of spiracles or a distal coronet of setae present (Fig. 13C).

Anterior trunk, thorax, consisting of pro-, meso-, and metathorax.

Prothorax with general outlines visible. Prothorax is heavily sclerotized; posterior part of the head capsule can be seen retracted into the prothorax (Figs. 13A–13C).

Mesothorax has two distinct pigment dots at the hind edge (Figs. 13A and 13C).

Metathorax has no spiracles (Figs. 13A–13C).

Posterior trunk (abdomen) with 8 units (Figs. 13A–13C).

Abdomen units 1–6 are bearing distinct lines of pigmentation, two medially on all tergites, and two laterally on most tergites (Figs. 13A–13C).

Abdomen unit 7 has no details preserved; only general outlines can be seen (Figs. 13A–13C).

Trunk end (Abdominal unit 8) with only general outline can be seen: the trunk end square in general shape, with a rounded posterior edge, when viewed in the dorsoventral view (Figs. 13A and 13B). No spiracles or anus can be discerned.

Discussion

Systematic interpretation

All specimens can easily be identified as larval forms of Diptera. This interpretation can be based on the general body shape of the specimen, the absence of walking (“ambulatory”) legs on the thorax, as well as the spiracle arrangement. The six morphotypes differ in numerous characters; their systematic relationships are discussed.

Morphotype 1: This morphotype is interpreted to be a representative of the group soldier and timber flies (Stratiomyomorpha) based on the following combination of characters (see Figs. 1A–1C, 2A–2C and 3A–3C; Figs. S1–S7): larva elongated and flattened, with head, thorax and eight abdominal units; body with oval pellets or scales, resembling calcium carbonate scales ; presence of such scales is a synapomorphy of Stratiomyidae+Xylomyidae (Figs. 2A–2C, 3A–3C and 4A–4C; Figs. S1–S7). The thorax of these specimens bears oval pellets or scales, rather than hardened sclerites as in Xylomyidae (Fig. 2B). Mandibles and maxillae are conjoined into a mandibular-maxillary complex (Figs. 4B and 4C). Larvae possess a large molar grinder and a setal brush at this mandibular-maxillary complex (Figs. 4B and 4C). The brush of the mandibular-maxillary complex, as indeed complex itself, is substantially reduced and simplified (Figs. 4B and 4C). This condition is, however, not uncommon among extant representatives of Stratiomyidae, for example in mature larvae of Hermeteiinae and Sarginae.

Despite the overall similarity with larvae of Stratiomyidae, the fossil larvae of morphotype 1 exhibit several traits unknown among any modern forms of Stratiomyomorpha in general. (1) The head capsule of the fossil larvae is extremely elongated with tentorial arms and metacephalic rod reaching back up to the posterior edge of the prothorax (Fig. 3B). (2) The larvae possess long triangular spines on the tergites of the trunk, as well as smaller rounded spines on the sternites of the trunk. This condition is unique among known larvae of Stratiomyomorpha and probably represents an autapomorphy of the morphotype (Fig. 3C).

This new morphotype clearly differs from two other types of stratiomyomorphan larvae recently reported from the Burmese amber (Liu et al., 2019) by the presence of the extremely long and strong spines on the trunk in the new form, as well as by the absence of the long setae on the posterior trunk (abdomen) and anterior trunk (thorax) (Figs. 2A–2C).

While the combination the features is, so far, unknown for Stratiomyomorpha, some of the characters are similar to the other larvae of Diptera. Extremely elongated head capsules and large tergal spines are known in larvae of Asiloidea, especially in the groups Mydidae and Bombyliidae (Marshall, 2012). An elongated metacephalic rod is in particular common in larvae of Mydidae, Xylophagidae, Thervidae and Scenopidae (Hennig, 1952; James, 1981; Irwin & Leneborg, 1981; Kelsey, 1981; Wilcox, 1981). This makes the interpretation of morphotype 1 larvae relatively challenging, due to the “chimaera-like” combination of the traits, as a probable result of the “push of the past” effect (Baranov, Schädel & Haug, 2019; Haug & Haug, 2019). This effect seems quite common among fossil representatives of Holometabola, representing phenomena occurring when initial diversification events in extant hyperdiverse groups lead to a number of “experimental” morphologies (Budd & Mann, 2018; Haug & Haug, 2019). In total we have found 13 larvae of this morphotype, with seven of them being preserved in just two amber pieces (four in PED-0031 and three in PED-0152). Almost all larvae (except PED-0031_1 and PED-0031_2) show signs of severe, most probably pre-mortem damage, such as squashing, full-body piercing, and splitting the body medially (along the pleural region). In some cases, we even see complete mutilation with entire parts of the body (i.e., thorax) being absent from some specimens. The high abundance of this morphotype, as well as their high incidence of damage indicates that these larvae were both frequent, and probably a preferred prey to the other inhabitants of the amber forest in Myanmar. We discuss further aspects of the ecology further below.

Morphotype 2: This morphotype is featuring prominent oval pellets or scales, similar to calcium carbonate nodules of modern larvae of Stratiomyidae. Therefore, we consider this morphotype as a likely ingroup within Stratiomyidae (Figs. 5A, 5B and 6A–6D; Figs. S9A and S9B). A further interpretation within Stratiomyidae is more challenging, due to the relatively poor preservation. Yet, the absence of a coronet of so-called “hydrofuge” setae on the terminal end and a relatively short body both point towards a terrestrial mode of life (McFadden, 1967). Yet such autecological generalizations should be approached with caution. There are species with terrestrial larvae in groups that otherwise have mostly aquatic larvae (e.g., Oxycera (Oxycera) leonina (Panzer & Wolfgang, 1798; Rozkošný, 1997). Also, the other way round, there are species with aquatic larvae in groups that generally have terrestrial larvae (e.g., Ptecticus; Jung et al., 2012). Therefore, morphology of the fossil alone can be an indication, but never a proof of the autecological affinities of an animal.

Also, this new morphotype clearly differs from two other types of Stratiomyomorpha larvae recently described from amber from Myanmar (Liu, Hakim & Huang, 2020), by the much longer head capsule (in relation to the body), absence of the any spines on the tergites, as well as by the absence of long setae on the trunk.

Morphotype 3: This morphotype clearly has closer relationships with Stratiomyidae based on the presence of a honeycomb pattern formed by oval pellets or scales, presence of a mandibular-maxillary complex and presence of brushes on this complex. Additionally, the habitus of the larva is highly reminiscent of extant terrestrial larvae of the group Stratiomyidae (see below). Within Stratiomyidae, the specimen can be interpreted as an ingroup of Pachygastrinae based on the following combination of characters: absence of a coronet of so-called “hydrofuge” setae on the trunk end; larva uniformly colored; trunk tergites with transversal rows of six setae each; labium not sclerotized and weakly developed; dorsal part of the mandibular-maxillary complex sclerotized; small larva, less than 10 mm (Rozkošný, 1982). Within Pachygastrinae the specimen appears most similar to larvae of the group Gowdeyana Curran, 1928 in lacking cuticular “teeth” along the anal opening; thoracic leg group setae paired; all setae in the dorsal transversal row are subequal (Figs. 7A, 7B and 9).

In general, the larva is relatively unusual for Pachygastrinae, as it is larger than most last-stage Pachygastrinae larvae (9.5 mm vs. 3–8 mm) and has a peculiar trunk end, elongated, ending with two large spines around the anus. It is possible that this larva belongs to an extinct lineage of Pachygastrinae, and large spines on the trunk and trunk end could represent an autapomorphy of this lineage. Yet, one should bear in mind that larvae for less than 10% of extant species of Pachygastrinae are known (Bucánková, Kovac & Rozkošný, 2009). Hence the possibility remains that the larva may also belong to an extant ingroup of which the larvae are not yet known.

Currently there are two species of Pachygastrinae known as adults from Dominican and Mexican amber: Pachygaster hymenaea Grund & Hauser (2005) (Figs. S11A and S11B) and P. antiqua James, 1971. The new fossil larva does not fit into the group Pachygaster, as in contrast to larvae of Pachygaster, the new larva does not have three setae in the “thoracic leg group” of setae. Also, the new larva is notably larger than any known larva of Pachygaster (Grund & Hauser, 2005). It is important to note however, that the specimen is rather poorly preserved, and identification should be seen rather as approximation of the phylogenetic affinity rather than final conclusion.

Morphotype 4: This morphotype clearly has closer relationships with Stratiomyidae based on the presence of a honeycomb pattern formed by oval pellets or scales, presence of a mandibular-maxillary complex and presence of brushes on this complex. Additionally, the habitus of the larva is highly reminiscent of extant terrestrial larvae of the group Stratiomyidae (see below). Within Stratiomyidae, further identification is impossible, due to the insufficient preservation of the specimen. The habitus in general is reminiscent of terrestrial larvae of Stratiomyidae, that is, from the ingroup Pachygastrinae.

Considering the exceptional preservation of this cuticle fossil, it is important to remember the possibility of contamination of the geological record by modern day holometabolan larvae, in particular fly larvae (Rasnitsyn, 2008). Fly larvae are known to crawl into narrow fissures within shales and other types of rocks, effectively creating a hard to spot contamination in the fossil record. The specimen in question has its cuticle interlaced with numerous grains of the sedimentary matrix. In this aspect it is similar to the contamination of late Cretaceous sandstone by an extant fly Protophormia terranovae, as reported by Rasnitsyn (2008: p. 249, figs. 96–97).

Yet, the fossil in general seems not to be entirely dissimilar from other euarthropodan fossils known from the same formation, in terms of its preservation (Gašparič & Hyžný, 2015). Additionally, the specimen was collected from a fresh split rock sample and an imprint was observed on the negative (unfortunately not collected). Still, we cannot entirely rule out that this larva is an extant contamination of the shale (Gašparič & Hyžný, 2015).

Morphotype 5: This morphotype seems to be a representative of Stratiomyidae, probably of the ingroups Stratiomyinae, Raphiocerinae or Nemotelinae, based on the presence of a coronet of “hydrofuge” setae (Pujol-Luz, De Xerez & Viana, 2004). The apical position of this coronet on the trunk end is not compatible with an interpretation as an ingroup of Nemotelinae (Hauser, Woodley & Fachin, 2017).

Not much more information could be gained from the larva, except that the “hydrofuge”setae coronet indicates an aquatic, rather than a terrestrial habitat of the animal.

Morphotype 6: This morphotype is represented by two very poorly preserved fossils; therefore, no definitive statement on its phylogenetic affinity can be made. Nevertheless, we decided to include it here, due to the overall similarities in the body shape and presence of the coloration patterns of cuticle, like those in larvae of the group Odontomyia or other representatives of Stratiomyidae (Fig. S14). For these reasons we think that it is prudent to consider this as a probable fossil of the Stratiomyidae, though there are no definite ways to further support this. This morphotype is too poorly preserved for any detailed systematic interpretation. Not much more information could be gained from the larvae, since the poorly preserved body falls into the “unknown” habitat category of the morphospace.

The fossil record of Stratiomyidae

Given the important role of larval dipterans, their numerous ecosystem functions and their often very specific association with certain microhabitats (Baranov, Schädel & Haug, 2019), their fossil records can provide a wealth of paleo-ecological information. Hence these new larval stratiomyomorphan specimens widen our understanding of the respective paleo-ecosystems from which they originated. Even on the adult side, representatives of Stratiomyidae are rare in the fossil record, with only 73 occurrences (specimens) having ever been recorded (according to PBDB, for the search parameters see Methods). This number is however excluding representatives of a unique, extinct group of flies with long proboscides (Zhangsolvidae), known from the Early Cretaceous of China and Brazil, as well as Late Cretaceous of Myanmar (Peñalver et al., 2015). These flies have emerged as important pollinators of the gymnosperm plants in the Cretaceous (Peñalver et al., 2015).

It is common for many organisms living in water to leave traces of their activity. Hence it should not be surprising that in the deep past ichnofossils provide most of the geological record of larval activity of Stratiomyidae, rather than body fossils. The most common example is the Jurassic “ichnogenus” Helminthopsis Heer, 1877. It was interpreted as originally caused by larvae of soldier flies of the group Stratiomys Geoffroy, 1762 or at least a closely related species (Mángano, Buatois & Claps, 1996). This expands the potential range of the geological record of the group from the Barremian (Cretaceous) to the mid Jurassic (Mángano, Buatois & Claps, 1996; Pickerill, Han & Jiang, 1998). Body fossils of stratiomyomorphans, as mentioned, are rare. All known larval fossil records are listed in Table 1, together with the material used in this contribution. Myanmar amber seems to be particularly rich in stratiomyomorphan larvae, as the number of morphotypes known from this deposit now has reached four. Liu, Hakim & Huang (2020) have described two morphotypes of stratiomyomorphan larvae from this amber. Both morphotypes are characterized by features intermediate between two stratiomyomorphan ingroups, Stratiomyidae and Xylomyidae (Liu, Hakim & Huang, 2020). Such chimera-like characteristics are also apparent in the one of the new morphotypes, namely morphotype 1. It can be interpreted as a result of the “Push of the past” phenomenon (Budd & Mann, 2018). In contrast to morphotype 1, morphotype 2 from Myanmar amber has a much less conspicuous morphology, and seemingly is a representative of Stratiomyidae s. str. as characterized by Hauser, Woodley & Fachin (2017).

The record from Dominican amber, morphotype 3, is only the third record of the group Stratiomyomorpha from this otherwise very productive deposit (Grund & Hauser, 2005). Only two specimens of the species Pachygaster hymenea Grund & Hauser, 2005 and a single specimen of Nothomyia sp. (Poinar & Poinar, 1999) has so far been reported from Dominican amber. This could indicate that representatives of Stratiomyidae were either very rare in the Miocene of Hispaniola, or alternatively their autecology was precluding them from being preserved in amber (Solórzano Kraemer et al., 2018).

The modern fauna of the isle of Hispaniola includes 13 species of Stratiomyidae (Perez-Gelabert, 2008). This relationship of fossil specimens to extant species is quite different from the situation with another ingroup of Diptera: Chironomidae (non-biting midges). For Chironomidae, there are more fossil species known from Dominican amber than there are extant species on the entire island of Hispaniola (Grund, 2006). The situation of Chironomidae in Dominican amber can be explained by the fact that more attention was given to fossils of Chironomidae of Hispaniola than to the extant ones. The same explanation cannot be applied to the discrepancy in species richness of fossil and extant species of Stratiomyidae. Soldier flies are mid-sized or even large flies; hence, they have a much lower chance of being overlooked in the amber records than Chironomidae. Modern representatives of Stratiomyidae in the Neotropics and other tropical regions are associated with open areas in the forests or forest canopy (Woodley, 2009; Hauser, Woodley & Fachin, 2017). We can therefore hypothesize that Dominican amber was capturing primarily animals associated with tree trunks, rather than canopy fauna or fauna of the open meadows within the forest. A similar capture pattern was shown for the Madagascar copal (Solórzano Kraemer et al., 2018).

The cuticle fossil from Činžat (morphotype 4) originates from deep-water, low energy sedimentary environment (Gašparič & Hyžný, 2015). It is difficult to explain how a larva of seemingly terrestrial Stratiomyidae ended up there. One possible explanation could be that the specimen drowned with driftwood and other terrestrial debris (which are present in the deposit) after a storm event.

The larva from Baltic amber is poorly preserved, and only identifiable as a larva of a soldier fly by the presence of the coronet of setae on the rear end. Overall, the fossil resembles extant aquatic larvae of the group Odontomyia (Figs. S14A and S14B), however, there are not enough diagnostic characters for a conclusive identification (also see “Discussion” above). It is still conceivable that this specimen represents a species of Stratiomyidae with an aquatic larva. There are several larval forms of Insecta that have an aquatic lifestyle and have been recorded from Baltic amber. This includes immature of Odonata (damselflies), Ephemeroptera (mayflies), Plecoptera (stoneflies) and Trichoptera (caddisflies) (Wichard, Gröhn & Seredszus, 2009). Martínez-Delclòs, Briggs & Peñalver (2004) suggested that aquatic larvae of Insecta can well be entrapped by still sticky resin pouring into water. This was probably the case for the larva from Baltic amber. This further supports that at least part of the Baltic amber deposits was formed directly next to water, probably in a swampy environment (Wichard, Gröhn & Seredszus, 2009).

The possible record of larvae of Stratiomyidae from the Eocene of the former maar Lake Messel might represent a rare find of aquatic insect larvae from this deposit. Unfortunately, the larvae are too poorly preserved for the detailed interpretation. Aquatic insects are generally rare in the oil shale of Messel and in other maar lake deposits, because the fossil-bearing sediments (the so-called oil shale) formed only in the deeper parts of the former maar lake, not in its shore-region (Wedmann, 2018). Aquatic insects, such as some larvae of Stratiomyidae were living in the shallow water in the shore region, and they could be only preserved as fossils when they drifted into the deeper, anoxic parts of the meromictic lake where the oil shale was formed.

Eco-morphological consideration

The ecomorphotype, or a shape of an organism adapted to a certain ecological condition, is used here as a proxy for the diversity of forms within a group of organisms (Haug et al., 2020b). Outlines of the entire body, or parts of it have been shown as superior proxies for the shape of an organism in many cases, when landmarks are hard to define, or when such landmarks do not reflect the shape of the organisms well enough (Tatsuta, Takahashi & Sakamaki, 2018). One of the most often used methods for the outline capture in geometric morphometry is a Fourier Elliptical Transformation (Tatsuta, Takahashi & Sakamaki, 2018; Polášek et al., 2018). This method allows accessing the diversity of ecomorphotypes of a group of organisms by examining body outlines. Here we used all available fossil and extant Stratiomyomorpha larvae to trace the changes in the larval morphospace occupancy of the group and, consequentially indirectly, ecological diversity throughout its history.

Ecomorphology of extant stratiomyomorphan larvae

New material examined in this study has shed a light on the far greater larval diversity of the group Stratiomyomorpha in deep time than was expected from the previously known geological records. We analyze the diversity of the ecological morphotypes of stratiomyomorphan larvae through time comparing it to modern ecomorphotype diversity. Here we use ecomorphotypes as a stable shape of an organism that evolved in response to certain ecological conditions (Rotheray, 2019).

Stratiomyomorphan larvae are occupying three main types of habitats: (1) aquatic, (2) terrestrial, mostly upper soil, leaf litter, and lower vegetation, and (3) living in wood, hence a saproxylic lifestyle. Many of the extant larvae of Stratiomyomorpha, in particular larvae of Pachygastrinae, are terrestrial saprotrophic and live under the bark of dead wood (McFadden, 1967; Marshall, 2012).

Larvae of Pantophthalmidae are saproxylic, inhabiting living wood (Marshall, 2012). Many other larvae of Stratiomyidae (i.e., not those of Pachygastrinae), are occupying aquatic habitats. In the fossil record, we have some indisputably aquatic larvae, that is, larvae Odontomyia sp. from Randecker Maar (Kühbander & Schleich, 1994) or larvae of Stratiomyia from the Holocene of Germany (Karl & Bellstedt, 1989). The original habitats of other fossil larvae are less clear (Whalley & Jarzembowski, 1985; Liu, Hakim & Huang, 2020).

We have attempted to compare ecomorphotypes of the extant aquatic, terrestrial and saproxylic stratiomyomorphan larvae with the morphotypes of the fossil larvae. In doing so we hoped to elucidate the changes in the stratiomyomorphan larval morphospace through deep time, as a response to the changing environmental conditions. Our analysis has shown that stratiomyomorphan larvae are showing essentially four main morphotypes: (1) elongated aquatic larvae, roughly circular to oval in the cross-section, as larvae of Stratiomyia, Oxycera, Odontomyia, (2) terrestrial and saproxylic larvae with spindle-shaped or cylindrical bodies (Figs. 14A and 14B). Analyses of the shape distribution in morphospace have shown that thickness of the body and shape of the body at the ends are determining separation of the morphotypes. These two characteristics of shape are of predominant importance, as they are making major contributions into the principal components (P.C.) 1 and 2. These two P.C.s are explaining 36.1% and 21.2% of the shape variability respectively (Figs. 14A and 14B). It is important to note however, that no significant separation between the morphotypes exist, as ascertained by a MANOVA test. P.C.1 and P.C.2 components have p > 0.05, when the type of the habitat is used as an independent variable for morphotypes separation. This is also not surprising as “aquatic” and “terrestrial” groups of the larvae are overlapping broadly in the general shape, and the “saproxylic” larvae morphotype is deeply nested in the “terrestrial” morphospace (Figs. 14A and 14B).

Figure 14 Ecomorphospace occupied by extant and fossil larvae of Stratiomyomorpha. Both plots presenting the same morphospace, split by the different grouping variables. Total captured variation = 75%; 62.1% at PC1 and 12.9 % at PC2.

(A) Morphospace split by the larval habitat: violet-saproxylic, blue-terrestrial, green -“unknown” (fossils), red-aquatic; (B) Morphospace split by the geological age/deposit of the larvae: blue-extant, red-Burmese amber, the rest of the deposits are represented by the single labeled dots.

Ecomorphology of fossil stratiomyomorphan larvae

The fossil larvae are widely distributed in the stratiomyomorphan morphospace (Figs. 14A and 14B). Most Cenozoic larvae (from Slovenian shale, Messel, and Baltic amber) fall within the area occupied by modern forms. Also, some of the Cretaceous forms fall within the area occupied by modern forms.

Morphotypes 2, 3, 4 as well as the larvae from Liu, Hakim & Huang (2020) firmly fell into the part of the morphospace occupied by modern terrestrial ecomorphotypes (Fig. 14A). The larva from Baltic amber plots into the “aquatic” type habitats, as did the specimens from morphotype 1, due to their elongated body (Fig. 14A). Despite that, we are hesitant to claim that morphotype 1 larvae are aquatic. Specimens of this morphotype are lacking the tell-tale characteristics of (most) extant aquatic stratiomyomorphan larvae, the coronet of “hydrofuge” setae (Rozkošný, 1997). Additionally, the extremely high abundance of morphotype 1 larvae (at least by the standards of the dipteran larvae in an amber deposit) can be explained by a possible close association with tree trunks. It is possible that these larvae lived under the bark of trees, as seen in many extant larvae of Stratiomyidae (McFadden, 1967; Marshall, 2012). It is well known, that organisms associated with tree trunks in the amber forests had a higher chance of being preserved in amber (Solórzano Kraemer et al., 2018). On top of that, a rich set of the syninclusions present in the amber piece PED-0031 together with morphotype 1 larvae is pointing towards the terrestrial environment (Fig. S2 and S3). Such syninclusions include: a mite, a possible scale insect, parts of other representatives of Insecta, a fly of the group Bibionomorpha, a beetle larva, a spider and a millipede. This strongly indicates a terrestrial environment for morphotype 1 larvae.

Only one of the morphotypes described by Liu et al. (2019) falls outside of the morphospace occupied by the extant Stratiomyomorphan larvae (Figs. 14A and 14B).

Our analysis has shown that the morphospace of stratiomyomorphan larvae has become significantly larger over time. Only a small part of the occupied area of the morphospace was lost, when we consider the general body shape. We think that increase in the morphospace size of the Stratiomyomorpha can be explained by the gradual diversification of the group from the Late Cretaceous onwards as it was shown by Wiegmann et al. (2011).

Conclusions

The fossil record of dipteran larvae and pupae is generally skewed towards abundant forms from low-energy sedimentary basins, such as lake environments (Rasnitsyn & Quicke, 2002). Therefore, groups with primarily aquatic immatures and high abundance, such as Chaoboridae and Chironomidae are over-represented in the fossil record (Rasnitsyn & Quicke, 2002; Zherikhin, Ponomarenko & Rasnitsyn, 2008). Aquatic larvae of other dipteran ingroups, while rare, have occasionally provided unprecedented insights into the evolution and paleoecology of the group (Whalley & Jarzembowski, 1985; Chen et al., 2014).

Terrestrial larvae of Diptera have been until recently considered extremely rare (Grimaldi & Engel, 2005). Recent works, however, have shown that certain groups of terrestrial dipteran larvae can be quite abundant, at least in amber (Baranov, Schädel & Haug, 2019). Therefore, it is not entirely surprising to find new immature representatives of Stratiomyidae in Cretaceous, Neogene and Paleogene ambers as well as in other types of fossil deposits. Further in-depth studies of amber and compression fossils collections will certainly lead to more new discoveries pertaining to larval biology of Stratiomyomorpha and other groups of Diptera.

Supplemental Information

Supplemental Information 1 Extant taxa analyzed in the morphometric analysis, with their habitat and origin of the image noted.

Click here for additional data file.

Supplemental Information 2 Dataset of the Stratiomyomorpha larvae outlines.

Outlines of the larvae used for the shape analysis with Elliptical Fourier Transformation in R package momocs (see description of the submitted data). Outlines are created based on the images from the papers listed in Table S1.

Click here for additional data file.

Supplemental Information 3 R code used.

Click here for additional data file.

Supplemental Information 4 All representatives of the morphotype 1 Stratiomyomorpha, covered in this article.

All specimens are for scale.

Click here for additional data file.

Supplemental Information 5 Syninclusions in the amber.

A) Amber piece PED-0031; B) Morphotype 1 larvae (PED-0031-3) Morphotype 1 larvae (Amber piece PED-0031); C) Morphotype 1 larvae (amber piece PED-0041); D) Morphotype 1 larvae (PED-0041), larva, habitus, ventro-lateral.ventrolateral. Legend: 1–beetle larva, 2–possible cuticle of the morphotype 1 larva, 3–morphotype larva, 4–morphotype 1 larva, 5–mite, 6–possible scale insect, 7-partial– part of a representative of Insecta inclusion, 8 -– fly of the group Bibionomorpha, 9 – beetle larva, 10- Aranea – spider, 11- Myriapoda. – millipede.

Click here for additional data file.

Supplemental Information 6 Syninclusions in the amber piece PED-0031.

A-B) Coleoptera representative larvae; C) Myriapoda; D) Possible representative of Collembola; E) Bibonomorpha fly of the group Bibionomorpha; F) Mite; G) Coleoptera,beetle adult; H) Possiblepossible scale insect.

Click here for additional data file.

Supplemental Information 7 Trunk end of the morphotype 1.

A) Specimen PED-0041; B) Specimen PED-0031-2; C) Specimen PED-0362 ; D) Specimen PED-0243.

Click here for additional data file.

Supplemental Information 8 Larvae of the morphotype 1.

A) Specimen PED-0349; B) Specimen PED-0362; C) First fragment of the specimen PED-0025; D) Second fragment of the specimen PED-0025.

Click here for additional data file.

Supplemental Information 9 Larvae of the morphotype 1 in the amber piece PED-0152.

A) Overview of the amber piece; B-D) Photos of the individual larvae in the piece. Photo D was taken using Keyence BZ-9000 fluorescence microscope. Legend: 1-4 - larvae of the morphotype 1.

Click here for additional data file.

Supplemental Information 10 Larva of the morphotype 1, specimen PED-0025.

A) Habitus, overview; B) Trunk’s end, enlarged, with tergal spines clearly visible.

Click here for additional data file.

Supplemental Information 11 Larva of morphotype 2.

A) Overview of the amber piece LACM ENT 366281, with 5 larvae; B) Habitus, ventral view of the morphotype 2 larva from the LACM ENT 366281.

Click here for additional data file.

Supplemental Information 12 Fossil Pachygastrinae larva, morphotype 3 (PED-462).

A) dorsal view; B) dorsal view, marked. Abbreviations: hc- head capsule, pt - prothorax, ms - mesothorax, mt -metathorax, a1-a7 - abdominal units 1-7, te - trunk’s end.

Click here for additional data file.

Supplemental Information 13 A) Fossil Pachygastrinae larva, morphotype 3, (PED-462), dorsal view; B) Pachygaster atra, dorsal view.

Abbreviations: hc- head capsule; as - anterior spiracle; ds-dorsal setae; ad -antero-dorsal setae; lt-lateral setae.

Click here for additional data file.

Supplemental Information 14 Pachygaster hymenea, Patrick Müller collection.

A) Habitus; B) Close-up photo of wing.

Click here for additional data file.

Supplemental Information 15 Pachygastrinae, larva, morphotype 4, PED-463.

A) habitus, ventro-lateral view, right-hand side; B) habitus, ventro-lateral view, left-hand side.

Click here for additional data file.

Supplemental Information 16 Pachygastrinae, larva, morphotype 4, PED-463, Scanning Electron Microscopy.

A). Habitus, ventro-lateral view; B) Head, ventrally; C) Posterior trunk, ventrally; D) Cuticle with Calcium carbonate pallets up close.

Click here for additional data file.

Supplemental Information 17 Extant Larva of Odontomyia sp. from the PED research collection.

A) Dorsal view; B) ventral view. Abbreviations: hc-head capsule, pt-prothorax; mt- metathorax; a1-a6 - abdominal units 1-6; te- trunk end.

Click here for additional data file.

Supplemental Information 18 Ped_0025 ebay receipt.

Click here for additional data file.

Supplemental Information 19 Ped_0041 ebay receipt.

Click here for additional data file.

Supplemental Information 20 Ped_0031 ebay receipt.

Click here for additional data file.

Supplemental Information 21 Ped_0152 ebay receipt.

Click here for additional data file.

Supplemental Information 22 Ped_0243 ebay receipt.

Click here for additional data file.

Supplemental Information 23 Ped_0349 ebay receipt.

Click here for additional data file.

Supplemental Information 24 Ped_0362 ebay receipt.

Click here for additional data file.

Supplemental Information 25 Ped_0113 ebay receipt.

Click here for additional data file.

Supplemental Information 26 Date of purchase of PED_0031 (29 March 2019).

Click here for additional data file.

V.B. is grateful to M. Spies and D. Doczkal (ZSM Munich) for his invaluable help with collection of ZSM as well as help with the literature. V.B. is grateful to B. Brown and G.-A. Kung (Los Angeles County Natural History Museum) for all the help during his visit to Los Angeles and work on the collection. We are grateful to E. McAlister for her help with digitizing specimens from the London Natural History Museum and Martin Hauser for his support and contribution to the manuscript improvement. We are grateful to the handling editor and three anonymous reviewers for their efforts in improving this manuscript. Thanks to all people providing free software. We are grateful to P. Müller and J. Damzen for their invaluable support. J.M. Starck and Carolin Haug, both LMU Munich are thanked for long standing support.

Additional Information and Declarations

Competing Interests

Author Contributions

Data Availability

The authors declare that they have no competing interests.

Viktor A. Baranov conceived and designed the experiments, performed the experiments, analyzed the data, prepared figures and/or tables, authored or reviewed drafts of the paper, and approved the final draft.

Yinan Wang conceived and designed the experiments, authored or reviewed drafts of the paper, analysis of the geological context, and approved the final draft.

Rok Gašparič conceived and designed the experiments, prepared figures and/or tables, authored or reviewed drafts of the paper, analysis of the geological context, and approved the final draft.

Sonja Wedmann conceived and designed the experiments, authored or reviewed drafts of the paper, analysis of the geological context, and approved the final draft.

Joachim T. Haug conceived and designed the experiments, performed the experiments, prepared figures and/or tables, authored or reviewed drafts of the paper, and approved the final draft.

The following information was supplied regarding data availability:

In accordance with guidance from the Society of Vertebrate Paleontology, PeerJ is not considering new submissions that deal with amber specimens purchased from sources in Myanmar after June 2017. This policy applies to all submissions received after the SVP letter was circulated to journals on 23 April 2020.

Although this article reports on amber specimens acquired in 2019, it was submitted to PeerJ on 20th December 2019 and received a first decision on 4th February 2020—therefore it was already under review before the policy was adopted. The authors have provided Supplemental Files detailing the acquisition of the specimens described herein.

Society of Vertebrate Paleontology: http://vertpaleo.org/GlobalPDFS/SVP-Letter-to-Editors-FINAL.aspx.

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
