# Peer review of "Eco-morphological diversity of larvae of soldier flies and their closest relatives in deep time"

_PeerJ, doi:10.7717/peerj.10356_

## Round 0.1 · original submission · Major Revisions

The reviewers have found your manuscript to be interesting, however, they have identified multiple issues with the submission. Please study their critique carefully and provide point-by-point response and the modified manuscript at the resubmission.

·

Basic reporting

Only a fraction of the literature is considered, several references in the text are not cited in the end. Sometimes there is a mixture between observation and conjecture.

Experimental design

fine

Validity of the findings

see main review

Additional comments

Diptera can be a challenging group to work with, being one of the four megadiverse orders. Working on fossil taxa, is much more complicated, because they have only a fraction of the characters discernable than recent specimens. Furthermore, the deeper in time, the more confusing the picture gets, and the closest living relatives might be on a different continent than from where the fossil was found. A solid understanding of the world fauna is therefore necessary to conduct research on extinct taxa. Especially cretaceous fossil can represent basal or extinct lineages, which makes their interpretation very complicated. Another challenge are immatures, which in Diptera have a completely different morphology than adults and only a fraction of the taxa are described, unfortunately often only in a superficial way, making this important field very challenging. The present paper deals with a very challenging field: fossil Diptera larvae. This is a very important contribution and I would like to see the descriptions of these fossils published, although I had some major problems with certain aspects of this paper, and therefore I don’t recommend publication but encourage resubmission.
I think it was a good decision of the authors not to name any of the taxa, because it is nearly impossible to place them in a meaningful genus (or species), and therefore only add names but not necessarily information to the system. Describing the morphology and discussing these fossils is more important than giving them names. Furthermore, I liked the approach of using morphospaces and novel techniques to analyze the fossils.
I have to admit that this was one of the more challenging reviews I have done in a long time and I was torn between being fascinated by the fossils and results and being annoyed about some of the sloppy and superficial aspects of the paper.
I try to give some constructive advice to aspects I am familiar with. Because I am not familiar with the morphometric methods used in the paper, I will not comment on their usefulness or applicability.

In general, the MS seems to be written relatively quickly and therefore shows many small errors, typos etc, which in themselves are not major, but are symptomatic for not spending enough time on the MS. For example, in line 164, listing the published sources of described larvae, the publications are in no order, neither chronological nor alphabetical, they include one reference (Dubrovsky 2004) which does not include a larval description, but at the same time the authors are missing a lot of the relevant publications. Of course, I think this is not the place to evaluate all described larvae, but major publications like Rozskosny (1981, 1982), Dusek & Rozkosny (1967), Schremmer (1951, 1986) Hennig (Die Larvenformen der Dipteren), Stubbs & Darke (2001) and some lesser known, but with very excellent larval drawings or pictures (Stuke 2003, Beuk 1990) were not consider. I am sure adding all these larval outlines into the analysis, might have influenced the results and therefore the discussion. Major catalog works like Woodley 2001, 2011, the catalog of fossil flies by Evenhuis 1994 are not mentioned, although they are relevant. It seems although the authors talk about Stratiomyomorpha, none of the larvae of Xylomyidae or Pantophthalmidae are used in the analysis and the Zhangsolvidae (for which no larva is known) are not even mentioned.
The lack of familiarity with the body of literature covering the immature stages of Stratiomyidae and related groups is one of my major points of critique. As a result, several mistakes crept into the MS, which could (and should) have been easily avoided: There are no predatory Stratiomyidae larvae, which gives the Stratiomyidae a unique biology amongst the Brachycera. Although I have observed Pachygastrinae larvae feeding on life insects, they might just be saprophytic and opportunistic when they encounter some insect they can overpower. Pachygastrinae are not feeding on the wood, they very likely feed on fungus and bacteria and decaying organic matter - so they are not saproxylic. In contrast to the xylophagus Pantophthalmidae, which have larvae boring through living (!) trees. The speculation about aquatic vs terrestrial larvae is difficult, because (as the authors mentioned) lack of a good phylogeny, but very likely all Stratiomyinae are monophyletic and aquatic, while the Nemotilini might be a group which independently developed aquatic larvae, and which is not part of the Stratiomyinae. The fact that some Oxycera larvae are terrestrial and some Ptecticus are aquatic might just be outliers, but worth mentioning.
I am not completely sure how the authors determine the morphotype, but maybe a key to the morphotypes would be helpful.
Concerning the descriptions:
1. A more technical aspect is the redundancy of the descriptions. It would be better to describe the overall morphology (what characterizes a Strat larva) in the beginning and not mentioning certain characters (or the lack of them) for every morphotype. For example, the fact that the mesothorax has no spiracles is mentioned in nearly every morphotype, while none of the Stratiomyidae larvae we know has spiracles in this segment. So why is it worth mentioning? And why is it not mentioned for morphotype 3? Same is true for the “posterior trunk” which “bears no apparent parapodia or creeping welts”. this is an uncommon character in this group and not worth mentioning. I would prefer if a bit more effort is made to describe the structure found in the fossils.
Similar to this aspect is that the authors did not differentiate what is observed and what is conjecture:
The statement that morphotype 5 is “probably metapneustic” is confusing, because the anterior spiracle is not discernable, so therefore it is more likely that it has an anterior spiracle like all other stratiomyomorpha, but which is not visible in the fossil instead of being the only metapneustic Strat larvae.
It is mentioned several times that the body of the fossil is covered in calcium carbonate cuticular scales, which is just an assumption, because they look like the scales of recent Stratiomyidae larvae, which have been chemically analyzed. There are similar looking structure on Ephydridae or Oestridae larvae, which are just thickened chitin.
The authors took in the descriptions a rather unique approach by trying to include crustacean terminology and coming up with a new system of body segmentation. This is not very helpful in two ways: first, the terminology for body parts is not even consistent between Diptera families, and experts of a family are used to a certain terminology, so this can be confusing and harder to compare with the older literature. Second, this can imply phylogenetic relationships of the structure, which are not always proven, like if the mouth-hooks of Diptera larvae are the same structure as the mandibles of decapoda. And writing “antenna [antennula]” does not really add anything to the paper or the description. Terms like “ocular segment” are very novel to the Diptera literature I am familiar with, and while I am not against better or new terminology, I did not see the deeper meaning in this. Especially having in the description: “Post-ocular segment 2 (intercalary segment) without externally recognizable structures” is not a real description, because this body part is not visible, so it is assumed by the authors that this structure should be there, but it is not seen in the specimen. This mixture of structures observed in the fossil and structures expected to be there are not a proper way to write a scientific paper. These structures should be discussed in the beginning, e.g. why the body has 20 segments, what is the post-ocular segment 5 etc. The authors mention they follow Borkent and Sinclair 2017 (another citation which is not in the references) for the terminology, but (if I am correct and this refers to the larval key in the Afrotropical manual) these terms are not included in this paper.

To summarize: The authors need to read and include more literature and incorporate more of the described larvae from the literature, especially Xylomyidae and Pantophthalmidae. They need to justify the unusual use of terminology, be more cautious to distinguish facts from conjecture, be more carful with formatting, citations and redundancy of the text.
This could be an important contribution after some major changes.


Some of the relevant literature:
Beuk, PLT 1990 Het voorkomen van Zabrachia tenella in Nederland. Entmologische Beitraege Amsterdam 50(8) 101-106
ROZKOSNRY., (1982): A biosystematic study of European Stratiomyidae (Diptera). Vol. 1. Introduction, Beridinae, Sarginae and Stratiomyinae. - Series Entomologica, 21: 1-401; The Hague.
ROZKOSNRY., (1983): A biosystematic study of European Stratiomyidae (Diptera). Vol. 2. Clitellariinae, Hermetiinae, Pachgasterinae and Bibliography. - Series Entomologica, 25: 1-431; The Hague.
STUBBS A, & DRAKE, M. (2001): British Soldierflies and their allies. - The British Entomological and Natural History Society, The Dorsett Press, Dorsett, 512 S.
Stuke 2003 Die Stratiomyidae und Xylomyidae Niedersachsens und Bremens, Braunschweiger Naturkundliche Schriften 6(4): 831-856

Reviewer 2 ·

Basic reporting

In their paper, Baranov et al. describe the diversity of Stratiomyomorpha larvae in the fossil record and report 23 new records belonging two six morphotypes. Given the scarcity of these larvae in the fossil record, the paper presents a very valuable addition to our understanding of the evolutionary history of Stratiomyomorpha. The manuscript is well-written, clearly structured and easy to follow.
The study includes a comprehensive background on the topic, which also describes the different types of extant Stratiomyomorpha larvae and their importance for modern ecosystems. The existing record of body fossils is reported as well. The description of the morphotypes is detailed and backed by convincing figures, which are all of high quality and among the best I know from similar studies on fossil insects. In particular, I like the approach to include colored versions of several photographs to highlight morphological features.

Experimental design

The scientific question is well-defined and relevant. All methods are clear and well described. The photographs, which are the basis of the main findings, are of very high quality.

Validity of the findings

The results are backed by convincing figures. All underlying data is provided.

Reviewer 3 ·

Basic reporting

no comments

Experimental design

no comments

Validity of the findings

no comments

Additional comments

The manuscript "Eco-morphological diversity of larvae of soldier flies and their closer relatives in deep time (#44242)" is a study with a focus on new records of fossil of the Stratiomyomorpha larvae from Cretaceous amber from Myanmar, Eocene Baltic amber, Miocene Dominican amber, and compression fossils from the Eocene of Messel (Germany) and the Miocene of Slovenia. However, I have several concerns about the manuscript.

I strongly suggest that the authors use specialized bibliography on immature insect forms, especially on Stratiomyidae, not on Crustacea! The appropriate literature influences the interpretation of the identification of the body parts of morphotypes and the taxonomic identification and terminology used in the work. The bibliography list below could help the authors to understand better some of the morphological characteristics observed in the described morphotypes. Especially, Figures 1 and 2 must not belong to the Stratiomyiomorpha group. The larvae of Stratiomyiomorpha (Startiomyidae + Xylomyidae sensu Pujol-Luz & Pujol-Luz 2014a) have a head with a cephalic capsule conspicuous. The mouthparts in Stratiomyidae and Xylomyidae are modified into a mandible-maxillar complex with modified brushes (Hennig, 1952; Schremmer, 1951). Mandibles are only present in the subfamily Chiromyzinae (Pujol-Luz & Pujol Luz, 2014b). Pantophthalmidae is an incerta sedis group outside of Stratiomyiomorpha group (Pujol-Luz & Pujol-Luz, 2014a, or infraorder), mainly because of the immature forms (larvae and puparia).

The larvae in Figures 1 and 2 do not have a head with conspicuous cephalic capsules, have no eyes or antennae, nor evidence of brushes. The mouthparts highlighted in the images (Figures 1 and 2) with the metacephalic rod developed, is similar to other modern families of Diptera larvae, but not to Stratiomyiomorpha (Stratiomyidae and Xylomyidae). The other morphotypes are certainly Stratiomyidae and should be described as such. They clearly show the larval cuticle encrusted with mosaic cells of calcium carbonate. Despite this, I do not believe it is possible to identify them at the subfamily level. For example, morphotype 5 was identified as Stratiomyinae subfamily, but it could have been perfectly identified as a Raphiocerinae subfamily (Pujol-luz et al, 2004). Therefore, I do not recommend the publication of the manuscript.

Cook, E. 1949. The evolution of the head in the larva of Diptera. Microentomology 14: 1-57.

Fusari, L.M, Dantas, G.P.S & Pinho, L.C. 2018. Order Diptera Thorp and Covich's Freshwater Invertebrates. 4ª Ed. Elsevier.

Godoi FSP; Pujol­-Luz JR. 2018. Family Stratiomyidae In Thorp and Covich's Freshwater Invertebrates. 4ª Ed. Elsevier. 771­777.

Hennig, W. 1952. Die Larvenformen der Diptera. 3. Teil. Akademie-Verlag, Berlin. 628pp.

Pujol-Luz JR, Pujol-Luz CVA 2014a. Pantophthalmidae. In: Roig-Juñent S, Claps LE, Morrone JJ (Eds) Biodiversidad de Artrópo¬dos Argentinos. Vol. 4. San Miguel de Tucumán, Editorial INSUE UNT, 391–397.

Pujol-Luz JR, Pujol-Luz CVA 2014b. Stratiomyidae. In: Roig-Juñent S, Claps LE, Morrone JJ (Eds) Biodiversidad de Artrópo¬dos Argentinos. Vol. 4. San Miguel de Tucumán, Editorial INSUE UNT, 396–406.

Pujol-Luz, J.R., Xerez, R. & Viana, G.G. 2004. Descrição do pupário de Raphiocera armata (Wiedemann) (Diptera, Stratiomyidae) da Ilha da Marambaia, Rio de Janeiro, Brasil.

Schremmer, F. 1951. Die -Mundteile der Brachycerenlarven und der Kopfbau der Larve von Stratiomys chamaeleon L. Osterreichische Zoologische Zeitschrift 3(3/4): 326-397.

Sinclair, B.J. 1992. A phylogenetic interpretation of the Brachycera (Diptera) based on the larval mandible and associated mouthpart structures. Systematic Entomology 17: 233-252.

Teskey, H.J, 1976. Diptera Larva associated with trees in North America. Memoirs of the Entomological Society of Canada. 53pp.

Reviewer 4 ·

Basic reporting

The article is fairly well written but does have some minor grammatical and typographic errors that will need to be addressed by the Editor.

Experimental design

I’m genuinely surprised these undescribed fossils were found on Ebay. However, what reassurance is provided by the sellers on ebay to confirm the true localities (and therefore ages) of these deposits? This should be addressed in the Materials and Methods.

Validity of the findings

no comment

Additional comments

The manuscript is a genuine contribution to the Stratiomyomorpha literature as the authors describe several new morphotype species from various fossils. The authors do well to describe the fossilised larvae fauna when even the extant taxa are relatively understudied. The figures are excellent, including a combination of high-resolution stacked photography and SEM that are well annotated. The authors also provide a thorough discussion of the systematic placement of theses fossils within the Stratiomyomorpha, along with comparison to previously published fossils.

I should note that I am not familiar with the morphology of larvae but extant adults, therefore I can only provide specific feedback regarding the general taxonomy of the world fauna. Saying that, I have provided feedback and comments where possible.

Introduction
Line 30: Any group above genus is referred to as in pleural, i.e. “Stratiomyomorpha are a group…”
Line 32: replace “ingroups” with “families”, i.e. “The major families are…”
A few minor formatting issues are present on Line 36 (space prior to fll stop) and Line 41 (extraneous hyphens) – please correct throughout.
Line 39: please replace “ingroup” with “subfamily” i.e. “Stratiomyinae (subfamily of Stratiomyidae)”
Lines 40 & 43: replace “groups” and “ingroups” with “subfamily”, i.e. Pachygastrinae, Clitellariinae, Hermetiinae and Sarginae.
In the second paragraph referring to Stratiomyomorpha larvae, can the authors please provide brief examples of the biology of the other families Xylomyidae and Pantophthalmidae?
Line 50: a revision of Hermetia illucens was recently published: Lessard B.D., Yeates D.K. & Woodley N.E. 2018. Revision of the Hermetiinae of Australia (Diptera: Stratiomyidae). Austral Entomology 58, 122–136 doi: 10.1111/aen.12333 (online publication date 16 April 2018).
Line 55: please provide the author of the genus on first mention, i.e. Stratiomys Geoffroy, 1762. Please do this throughout the manuscript for all genera mentioned for the first time. Thisis important for taxonomic papers.
Line 56: replace “group” with “genus”, i.e. “genus Stratiomys”. Where possible, be specific to taxonomic rank.
Line 58: Please clarify which specific group the authors refer too, i.e. Helminthopsis?
Line 61: replace “records” with “studies” or “deposits with fossilised Stratiomyomorpha larvae” as there are more than five recorded morphotypes from the deposits listed below.
Line 75: please insert references to fossil Odontomyia larvae. Also italicise the genus name here.
In this paragraph can you please add the year range (MY) for each deposit, as this is information is missing from the Burmese amber, Eastern German and Isle of Wright deposits, but provided for the Lerida and Randecker Maar deposits.
Line 76: please elaborate on why larval forms are crucial for Holometabola, or state explicitly that the exploitation of various aquatic and terrestrial substrates has potentially led to their hyper diversification, including beetles, wasps, etc.
Line 80: Can authors state how many new morphospecies are described in the paper. It might be nice to also include a brief summary of the depositions where these new fossils were found, i.e. “… based on new fossil specimens found in deposits including Burmese amber, xyz”.
There should be a discussion on the larval autapomorphic morphological characters that defined the Stratiomyomorpha (i.e. see Line 568 from Discussion).

Results
Morphotype 1 – should the mite be mentioned in a Remarks section?
Discussion
Line 680: capitalise Table 1.
Line 686: correct “morphotype.s” and similar typographic errors throughout manuscript
Line 687: explain in text ‘push of the past’ reference.
Line 790: Inopus is mispelt
Line 805: remove “(xxx)” or replace with reference.

Figures
Scale bar of Figure 10 is difficult to read – perhaps place a white background behind it as in other figures.
There is no figure legend provided in Figure 11.

---

## Round 0.2 · Minor Revisions

Thank you for significantly improving the manuscript. However, there are several things that need to be fixed before I can recommend this manuscript for acceptance. First of all, please make sure that your citations are appropriate and correctly formatted. Second, please read the feedback by reviewer 1, where (s)he discusses the issue of phylogeny reconstruction. This is an extremely valuable and valid point that should be addressed in your manuscript.

·

Basic reporting

no comment

Experimental design

no comment

Validity of the findings

no comment

Additional comments

I am very glad to see that the authors added new literature and did some recommended changes, as well as discussed all the points and had satisfying answers to each point. I am looking forward to seeing the work published, it has in my eyes a rather unconventional approach and therefore adds to the scientific body new aspects, which might inspire more research on fossils and larvae, which is much needed. Also, the authors drew more general conclusions about the change in the “morphospace” of this group, which is significant beyond the group of Diptera investigated.

A few comments:

- It is always irritating to encounter new terminology from the one somebody is used to, this is just a human condition that we resist change. So reading crustacean terminology is unusual for a Dipterist, but the idea that Diptera are just a small part of the crustacean lineage is correct, like birds are dinosaurs etc. Of course many Ornithologist would be confused when their books would use paleontological terminology and when the bird books would focus on skeletal characters instead of feathers (you can t see the bones in a flying bird, but there are there). And yes, it would help to unify the terminology, so I leave it to the authors to use whatever terminology they want to use, especially because they illustrated and explained sufficiently what they use and what they mean with these terms.
- Of course, the authors are correct that “nobody can prove the phylogenetic background”. What can be done is to test a hypothesis, and with the help of concepts developed by Hennig and others, we can test the hypothesis of homology and falsify these according to Popper. It has been extremely challenging in the past (during efforts for the “Fly-tree”) to establish the homology between different types of Diptera larvae, therefore I am careful in establishing homologies over a much greater evolutionary distance. But this is not impossible and it is up to the authors to present their hypothesis.
- Yes, there are things we “don’t see” but they “are there” like the intercalary segment. But my point was that we must differentiate between what we see and what is there. If there is only half the larva preserved as a fossil, it is reasonable to assume that this larva has the same number of segments as all the others, but for the description we must only mention what is there. In the discuss we can mention that this specimen likely had all the segments etc… So yes, the information needs to be included, but it should not be mixed with the “discussion/interpretation”.
- The citation “The morphological terminology largely follows Borkent and Sinclair (2017).” Is still there and my point: “The authors mention they follow Borkent and Sinclair 2017 (another citation which is not in the references) for the terminology, but (if I am correct and this refers to the larval key in the Afrotropical manual) these terms are not included in this paper” was not addressed.
- The Afrotropical Manual is cited as: “Hauser, Martin, Norman E. Woodley, and Diego A. Fachin. 2017. “Stratiomyidae (Soldier Flies).” In Manual of Afrotropical Diptera. Volume 2: Nematocerous Diptera and Lower Brachycera. 633-640, edited by Art Borkent and Bradley J. Sinclair, 2:919–79. Suricata 5. Pretoria: South African National Biodiversity Institute.” But it is not edited by Art Borkent and it should be cited (according to the Manual itself) as: “Hauser, Martin, Norman E. Woodley, and Diego A. Fachin. 2017. “Stratiomyidae (Soldier Flies).” In: Kirk-Spriggs, A.H. & Sinclair, B.J. (eds), Manual of Afrotropical Diptera. Volume 2: Nematocerous Diptera and Lower Brachycera. Suricata 5. Pretoria: South African National Biodiversity Institute. 919–79.
- The reference of Woodley 2011 is not of significant importance for this paper. If the authors would have contacted me, I could have provided a PDF of this work.

Reviewer 3 ·

Basic reporting

No comments.

Experimental design

No comments.

Validity of the findings

No comments.

Additional comments

No comments.

---

## Round 0.3 · Minor Revisions

You have substantially improved the manuscript. However, there are numerous typos, missed articles, verbs, etc. Please make sure that a native speaker or a technical editor works on your paper before resubmission. I have edited the first 7 pages, but some paragraphs require serious re-writing for clarity.

---

## Round 0.4 · Minor Revisions

Overall, the manuscript was improved a lot. However, there are consistent issues with grammar. Articles and verbs (is, was, are) are frequently missing, Please make sure that your manuscript is carefully checked before resubmission. I have edited the first several pages, to show my main suggestions.

---

## Round 0.5 · Minor Revisions

Thank you for making the revisions. The submission is almost ready to be accepted.

As agreed with the Section Editors, please address the following:

1. Please be more specific in the manuscript about the source and dates of purchase/accession of the material

2. Please include (as supplemental material) some evidence of how you obtained the specimens - for instance, the eBay records or similar. These would be made available alongside the published article.

The published article will be accompanied by a statement from the journal that reiterates our support for the moratorium requested by SVP but acknowledges that the article was written and submitted prior to the SVP letter being circulated.

---

## Round 0.6 · accepted · Accept

Thank you very much for making the suggested changes. The paper is now in acceptable shape, and I recommend it for publication.